**Perspective**

# Harnessing the structural evolution of metal–organic frameworks under electrocatalytic conditions
Zheao Huang ⓘ ✉ & Dominik Eder ⓘ ✉

Metal–organic frameworks (MOFs) are often dismissed in electrocatalysis due to their structural "instability" under operating conditions. In this Perspective, we reframe MOF evolution in electrocatalytic conditions as a controllable pathway for accessing highly active catalytic species, such as metal (oxy)hydroxides, surface defects, and open metal centers. We highlight how leveraging structural evolution, rather than complete degradation, can be harnessed through rational design and activation strategies. Operando/in-situ techniques are highlighted as essential tools for tracking in-situ structural dynamics and associated evolution mechanisms. By integrating these design, characterization, and modeling insights, this Perspective outlines a framework for turning structural evolution into a powerful tool for catalytic functionality.

Metal–organic Frameworks (MOFs) are crystalline structures formed via the self-assembly of metal nodes (or metal cluster) and organic ligands, often creating a periodic porous network framework[1,2]. Over the past decade, MOFs and their derivatives have been extensively explored in electrochemical applications, owing to their tunable pore architectures, high surface areas and the molecular proximity of modifiable inorganic and organic building blocks, which enhance charge separation and transfer while offering a high density of catalytic redox sites[3,4]. However, MOFs often struggle to maintain long-term structural stability under harsh electrocatalytic conditions, which is attributed to the relatively weak coordination bonds between metal nodes and organic ligands compared to the stronger chemical bonds found in purely inorganic materials[5]. In particular, under strongly alkaline conditions, such as in the widely used 1 M KOH electrolyte, MOFs are prone to ligand exchange, where hydroxide ions readily substitute the original ligands. Another degradation pathway is "corrosion", which is exacerbated under high anodic potential required for reactions like the oxygen evolution reaction (OER), where the ligands are directly oxidized[6]. Additionally, phenomena such as metal leaching from the MOF framework may also occur[7]. These processes can lead to the partial or complete breakdown of the metal–ligand coordination in the MOF framework, raising concerns about their long-term structural stability and feasibility as direct electrocatalysts.

MOFs used in electrocatalysis can be broadly categorized into three classes: precursor catalysts, direct catalysts, and pre-catalysts[5,8,9]. Precursor catalysts refer to MOFs that are deliberately converted, e.g. via pyrolysis, sulfurization, phosphidation, or other treatments, into inorganic composites comprising small metal or metal oxide species dispersed within a carbonaceous matrix, prior to catalysis[10–12]. Compared to large macrocyclic molecules like porphyrins or phthalocyanines, MOFs offer the coordination interactions between the periodically arranged ligands can act as fences, facilitating better dispersion of the target metal atoms or metal oxide clusters on the substrate[13]. For example, Zhang et al. thermally activated a Fe-doped zeolitic imidazolate framework (ZIF) to embed atomically dispersed $FeN_4$ active sites into porous carbon, avoiding metal aggregation[14]. Similarly, Wang et al. exploited the strong coordination between $Ru^{3+}$ ions and the free amine groups ($-NH_2$) at the skeleton of UiO-66 to successfully anchor Ru single atoms onto a carbon substrate via pyrolysis[15]. Such isolated single-atom sites can maximize atomic utilization efficiency and enhance catalytic performance. However, this often leads to significant losses in surface area and porosity, as well as disruption of the intrinsic hybrid nature of MOFs, thereby undermining their key advantages in facilitating efficient charge separation and transfer between inorganic nodes and organic ligands.

Direct catalysts refer to MOFs that are used directly in reactions without prior treatments, with their structural integrity verified by comparing pre- and post-reaction characterizations, often supplemented by in-situ techniques. Examples include S-doped NiBDC nanosheets[16] and missing-ligand layered-pillared CoBDC[17]. Zr-based MOFs, such as NU-1000, also represent an important class of direct catalysts, exhibiting great chemical stability in water under neutral and even acidic pH conditions[18,19]. These studies aim to optimize the electronic structure of MOFs while maintaining their framework under specific conditions to achieve catalytic activity. However, such stability must be validated on a case-by-case basis for each MOF and reaction type, such as Zr-based MOFs fail in strong alkaline media, making them far less versatile than conventional metal oxide/sulfide catalysts[20].

Institute of Materials Chemistry, Technische Universität Wien, Vienna, Austria. ✉e-mail: zheao.huang@tuwien.ac.at; dominik.eder@tuwien.ac.at

Pre-catalysts refer to MOFs that undergo structural changes, ranging from subtle to substantial, during exposure to an electrochemical environment and/or throughout the electrocatalytic reaction. In the literature, a variety of terms have been used to describe this process such structural evolution, structural reconstruction, structural transformation, and structural degradation[21]. Before discussing the effects of these changes in MOF, it is important to clarify these definitions to avoid conceptual ambiguity. Structural evolution refers to the progressive changes in MOF structures during electrocatalysis and is generally beneficial, serving as an umbrella term encompassing reconstruction, transformation, and degradation. More recently, it has also been used to describe structural dynamics at the atomic level, particularly involving single atoms or open metal sites in catalysis[22,23]. Structural reconstruction involves the rearrangement of atoms or bonds, typically on the surface or throughout the framework, triggered by electrochemical activation. Structural transformation describes an irreversible shift to a new stable phase, often accompanied by phase transitions or the formation of entirely different crystalline or amorphous materials. Structural degradation, by contrast, entails the breakdown or collapse of the MOF framework, which is typically irreversible and detrimental.

While the concept of pre-catalyst and associated terms are not unique to MOFs and is also well established for transition-metal oxides, hydroxides, and chalcogenides, both can undergo electrochemically induced structural evolution to generate catalytically active species[24,25]. The key distinction lies in the presence of organic ligands in MOFs, which not only provide tunable coordination and porosity but also introduce additional degradation pathways such as ligand detachment or framework collapse that are absent in purely inorganic oxides. Moreover, the weaker coordination bonds in MOFs can facilitate more pronounced atomic rearrangements or phase transformations under operating conditions, whereas oxide pre-catalysts often retain part of their crystalline lattice as a structural backbone[5].

In this Perspective, we use the umbrella term "*structural evolution*" to represent all types of structure-related changes in electrochemical instability of MOFs, thereby avoiding confusion caused by inconsistent nomenclature. The potential structural evolution of such MOF catalysts often leads to the in-situ formation of new phases under the reactions, thereby altering the geometric and electronic structure of the metal nodes[26,27]. Therefore, before evaluating the MOF electrocatalytic performance, one must ask: *Does the MOF itself function as the true catalyst, or does it merely act as a pre-catalyst that evolves into the active phase under operating conditions?*

In general, MOFs can be functionally modified to enhance their catalytic activity/stability without altering their crystalline structure, or even

their electronic properties, through strategies such as ligand functionalization or metal doping[28]. In some cases, it is even possible to design highly conductive 2D MOFs[29]. This raises the possibility that certain MOFs may indeed serve as true catalysts, directly participating in electrocatalytic reactions. However, using MOFs as direct catalysts under operational conditions remains a subject of debate because subtle structural changes occurring during the reactions probable escape detection due to limitations in standard post-reaction characterizations, as discussed in detail by Zheng et al.[5] Therefore, given the various challenges in developing MOFs as direct electrocatalysts, increasing attention has recently shifted toward deliberately exploiting the structural "instability" by using MOFs as pre-catalysts, rather than pursuing the traditionally idealized notion of a "perfectly stable" MOF during catalysis[27]. The focus in the community has recently shifted to investigate structural evolutions in MOFs and are now gradually recognized as opportunities for performance enhancement rather than drawbacks. In this Perspective, we investigate the key factors influencing their structural evolution during electrocatalysis and proposes operando/in-situ techniques for real-time monitoring, along with strategic approaches to harness the structural evolution process (Fig. 1).

## MOF stability in electrolytes

In electrocatalysis, the electrolyte not only serves as an ionic conductor and reactant source but also strongly affects catalytic performance. To enhance activity, strong acidic or alkaline electrolytes are commonly employed as the aqueous solution, because they can provide high ionic conductivity due to the abundance of protons ($H^+$) and hydroxide ions ($OH^-$), acting as direct reactants to accelerate reaction kinetics[30]. Therefore, before evaluating electrocatalytic reactions, it is essential to assess not only the stability of MOFs in aqueous media but also their durability as catalysts under acidic and alkaline electrolytes.

Compared with neutral water, $H^+$ and $OH^-$ exhibit much stronger destructive effects on MOFs by competing with ligands for coordination to metal centers, ultimately leading to framework degradation[31,32]. Strengthening metal–ligand bonds is thus a key strategy to improve MOF stability in electrolytes. Both the charge density of the metal ion and the hydrophobicity of the ligand significantly influence the robustness of coordination bonds[31]. For example, MIL-101 with $Cr^{3+}$–O coordination developed by Leus et al. exhibited excellent stability for over two months in aqueous solutions across pH 0–12[33]. Similar metal strategies have been applied to MOFs based on high-valence $Zr^{4+}$, $Fe^{3+}$, and $Al^{3+}$ nodes[34–36]. On the ligand side, introducing hydrophobic groups can provide steric protection to metal sites. Zhong

**Fig. 1 | MOF structural evolution under electrocatalytic conditions.** Schematic representation of MOF structural evolution under electrocatalytic conditions, accompanied by operando/in-situ techniques and rational design strategies.

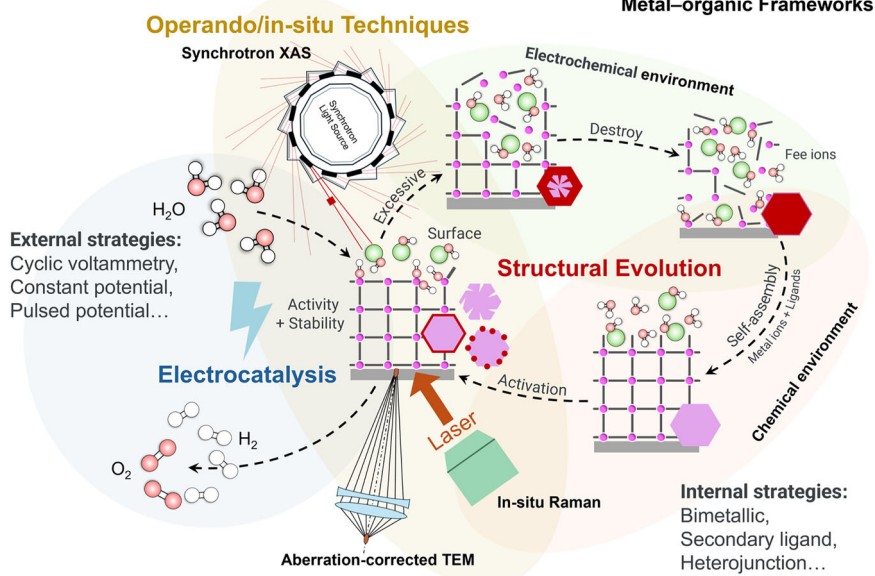

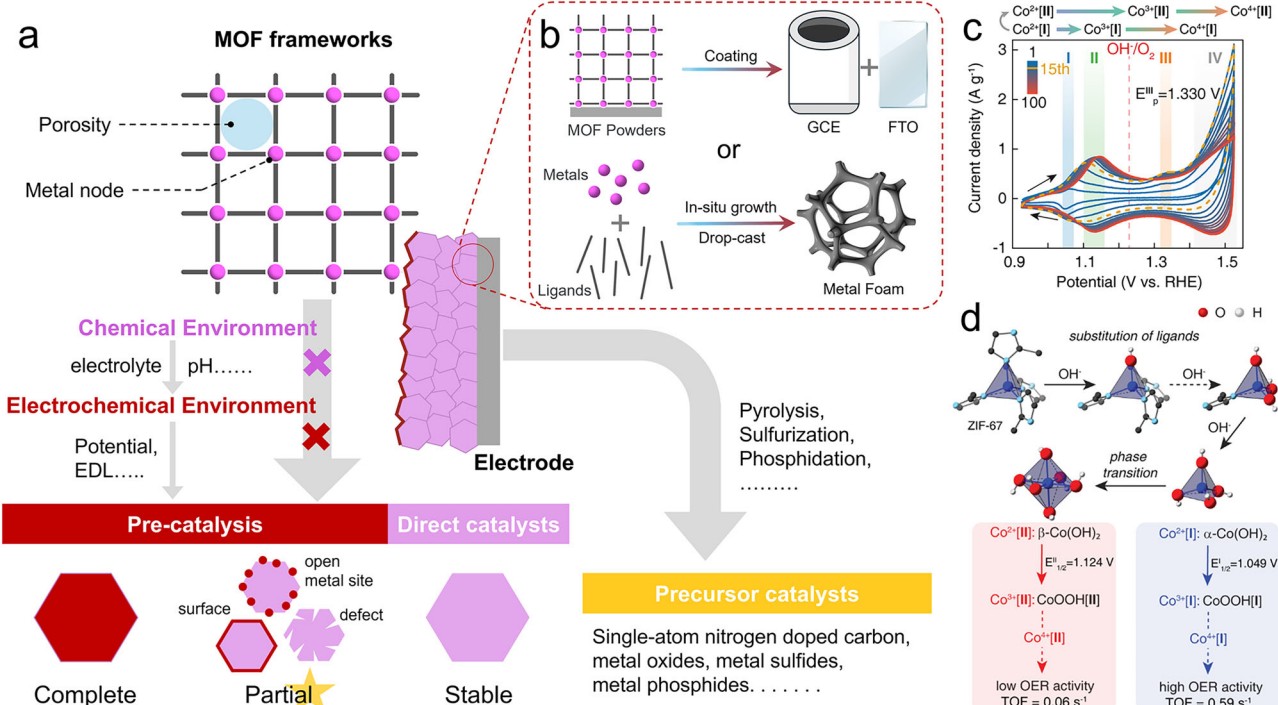

**Fig. 2 | Classification, electrode preparation, and precatalytic evolution.**
**a** Classification of MOFs as electrocatalysts, associated treatment methods, and
potential factors influencing MOF evolution pathways. **b** Selection and preparation
of MOF working electrodes in a typical three-electrode system. **c** 100 CV cycles of the
ZIF-67 between 0.925 and 1.525 V vs. reversible hydrogen electrode (RHE)[6].
**d** Precatalytic evolution of ZIF-67 to α- and β-Co(OH)$_2$ and their further oxidation
and OER activity[6]. Copyright 2019 American Chemical Society.

et al., for instance, developed pH-stable UiO-66 (Zr) variants by tuning the
position of trifluoromethyl substituents[37]. However, hydrophobic mod-
ification may also hinder the approach of electrolyte molecules to MOF
catalytic sites, which, while enhancing stability, could slow catalytic reac-
tions by limiting reactant accessibility. Post-synthetic modifications have
also proven effective: Liu et al. transformed PCN-426 (Mg) into robust Fe-
and Cr-MOFs using postsynthetic metathesis and oxidation strategies,
achieving stability in both strong acids and bases due to the formation of
inert $Fe^{3+}$–O and $Cr^{3+}$–O bonds[38].

Currently, although numerous reports claim pH-stable MOFs, most
rely solely on powder X-Ray diffraction (PXRD) patterns as evidence. More
rigorous validation should involve porosity analyses (e.g., $N_2$ physisorption)
and even operando/in-situ characterizations under electrolyte conditions.
For electrocatalysis, identifying or designing MOFs that remain stable across
a wide or targeted pH range is crucial, as this minimizes extrinsic inter-
ference and enables reliable interpretation of structural evolution and cor-
responding catalytic mechanisms.

## MOF electrode stability

Before delving into the electrocatalytic behavior of MOFs, it is essential to
clarify how the choice of MOF electrode and its corresponding preparation
method can influence performance. Typically, electrocatalytic testing is
conducted in a three-electrode system, which inevitably requires integrating
the MOF with the working electrode. While much attention has been paid to
the influence of the reaction environment on the structural stability of
MOFs, the stability of MOFs on the working electrode (i.e., electrode sta-
bility) is equally important but often overlooked. In some cases, even when
the MOF itself exhibits great chemical stability, poor adhesion to the elec-
trode caused by suboptimal electrode fabrication methods or binder issues
can lead to detachment of MOF particles into the electrolyte. This ultimately
results in diminished catalytic activity and complicates the interpretation of
reaction mechanisms.

To ensure reliable mechanistic insights, it is essential to distinguish
between MOF stability and electrode stability, and to develop well-

integrated MOF-based electrodes with long-term reaction robustness. A
wide range of working electrode substrates are available for electrocatalysis,
including carbon-based (cloth, felt), metal-based (nickel or copper foam),
glass-based (such as GCE and FTO/ITO), and thin-film electrodes[39]. These
differ in conductivity, surface morphology, area, and porosity, all of which
significantly affect the stability of the electrode and even the resulting cur-
rent output exhibited by the MOF.

Glass-based electrodes like glassy carbon electrodes (GCE) or
fluorine/indium-doped tin oxide (FTO/ITO) are commonly used with
drop-casting or spin-coating involving a Nafion binder to immobilize
MOF powders on flat 2D glass surfaces (Fig. 2b). This approach offers
uniform coatings ideal for photoelectrochemical applications, and more
precise operando/in-situ spectroscopic analysis. However, these planar
electrodes generally exhibit lower intrinsic catalytic activity compared to
metallic substrates, such as the widely used nickel foam (NF), which
provides great conductivity, 3D interconnected network, and hierarchical
porous structure[40,41]. Based on this, many recent studies employ in-situ
growth of MOFs on NF skeletons, aiming to enhance both electrode
stability and catalytic activity (Fig. 2b)[42–44]. This approach eliminates the
need for Nafion, thereby minimizing overpotential losses caused by
active site blockage, ion diffusion inhibition, and interfacial resistance[45,46].
However, it is important to note that MOFs grown directly on NF may
differ in morphology, crystallinity, or electronic structure from separately
synthesized MOF powders. This distinction is often overlooked in
research articles. Therefore, when using MOF@NF electrode, it is advi-
sable to perform additional characterization and comparisons with MOF
powders to ensure consistent understanding of material properties.

One additional limitation is that 3D porous structures like NF or
carbon cloth often pose challenges for in-situ spectroscopic techniques,
especially Raman spectroscopy, as the laser source struggles to reflect effi-
ciently through the porous matrix. In addition, the presence of extra metal
from the metal-based electrode interferes with the identification and cata-
lytic analysis of the intrinsic metal active sites of MOFs, especially using in-
situ growth methods.

In summary, before electrocatalytic testing, it is highly recommended to thoroughly assess the MOF electrode stability, and select an electrode substrate that not only supports effective MOF loading but also ensures long-term structural integration under the electrochemical environment.

## Chemical environment and electrochemical environment

The electrochemical environment refers to the chemistry of the electrolyte, e.g. solvents, pH, any coexisting ions, and the generated products of catalytic reactions[5]. The electrochemical environment addresses the effects of continuously or intermittently introducing applied potential through instrumental means, such as using an electrochemical workstation. The chemical and electrochemical environments, as external factors, critically affect the structural integrity of MOFs under electrocatalytic conditions (Fig. 2a).

A proper evaluation of MOF electrocatalytic performance must begin with an assessment of its stability under the relevant chemical environment. However, many studies tend to bypass this step, subjecting MOFs directly to electrocatalytic testing without evaluating their response to the chemical environment alone. This can lead to ambiguous interpretations of the potential structural evolution: *when changes occur, are they caused by the electrolyte, the applied potential, or are both responsible?* Without a clear answer, it becomes difficult to pinpoint the precise onset of structural evolution or to unravel the true catalytic mechanism. To avoid such complications, one must decouple the effects of the chemical environment from those of the electrochemical process. This requires ensuring that the as-synthesized MOF exhibits long-term stability in the chemical environment prior to any electrochemical testing.

Chemical stability here primarily refers to the MOF's resistance to degradation in various electrolytes, whether neutral aqueous solutions, acidic or alkaline media, or complex ionic matrices such as seawater[47–49]. The standard approach for evaluating chemical stability involves exposing the MOF to these environments for extended periods under open-circuit potential (OCP) or without applying an external potential. By comparing the structural, porosity, and morphological features of the MOF before and after exposure using basic characterization techniques, one can determine the maximum duration the framework remains intact under specific chemical environments[48]. Typically, this time period can be used as a benchmark for assessing the chemical stability of a given MOF only if the XRD peaks remain unchanged (with intensity loss below 5%), metal leaching is low, and porosity is retained.

In general, the choice of metal nodes and ligands following the Hard Soft Acids Bases (HSAB) principle, such as pairing hard acids with hard bases or soft acids with soft bases, yields more robust stronger interaction in MOF frameworks[50]. For instance, zeolitic imidazolate frameworks (e.g., ZIF-8, ZIF-67), constructed from $Zn^{2+}/Co^{2+}$ (soft acid) and imidazolate ligands (soft bases), exhibit great chemical and physical stability[51]. This also applies to many carboxylate-based MOFs constructed from metal-oxo clusters such as $Ti^{4+}/Zr^{4+}$ (hard acid) and carboxylate ligands (hard base). Recent data-driven approaches have furthered this understanding by predicting MOF stability in chemical environments using machine learning (ML) trained on large structural databases. For instance, Terrones et al. developed ML models using the WS24 dataset comprising over ~10,000 MOF structures to successfully predict hydrolytic stability[47].

Once the chemical stability of MOF is established, attention must turn to the electrochemical environment. When a potential is applied to the MOF electrode, especially under continuous positive and negative potential sweeps, an electrical double layer (EDL) forms at the electrode–electrolyte interface, which attracting ions of the opposite charges to accumulate on the MOF surface[52]. This can promote $OH^-$ migration and altering the local pH, ultimately leading to nucleophilic attack on the metal–ligand coordination bonds, ion/ligand exchange, and direct reduction/oxidation[27,53]. A common example is the electro-oxidation of transitions metal-based MOFs into metal (oxy)hydroxides[54–57]. And other, Heidary observed that the porphyrin-carboxylate ligands in Mn-MOF were cleaved and released under certain potentials[58].

Therefore, the theoretical basis for chemical stability often becomes insufficient under electrochemical environment, because the EDL alters the local chemical microenvironment at the MOF surface. This further emphasizes the importance of operando/in-situ electrochemical techniques for real-time monitoring of MOF structures and verifying potential structural changes, especially at the MOF surface. Of course, the criteria previously used to assess chemical stability can also be applied to evaluate electrochemical stability, but only as a complementary approach by comparing the intrinsic MOF properties and its diffraction patterns before and after the reaction. In addition, the electrochemical stability and associated structural evolution can often be rapidly assessed through cyclic voltammetry (CV) analysis. The positions and integrated areas of oxidation peaks in the CV curves can respectively indicate the presence of metal species in different oxidation states and the number of their electron-accessible sites, such as the various oxidation states of Ni and Co species (Fig. 2c)[6,59–61]. Researchers also commonly evaluate the catalytic activity/stability of MOFs by comparing current density and reaction time in long-term chronoamperometry (CA), often in combination with post-reaction characterizations[16].

## MOF structural evolution under electrocatalytic conditions

Under operating conditions, MOFs tend to undergo some degree of structural evolution in their electrochemical behavior, even during the initial activation stage, due to the relatively weak coordination bonds (M–L) and the chemical fragility of the organic components. Recent studies have increasingly focused on and strategically utilized this MOF evolution, as it often results in the generation of morphological changes, the introduction of defect-sites, and the formation of new catalytically active species. Examples such as high-valent metal (oxy)hydroxides[62,63], oxygen vacancies[64], and missing-ligand/cluster defects[17,65] have now been widely recognized. These features can effectively optimize the geometric and electronic structure of MOFs, such as unsaturated coordination environments, charge distribution/transfer, band gap and the density of electronic states near the Fermi level, thereby enhancing catalytic activity, as showed in the Reviews[27,66]. Such as, Shi et al. obtained oxygen defects ($M-OOH_v$) on the surface of $Fe_2Co$-MOF through in-situ evolution, which effectively regulated the electronic density of states and served as the real active sites for OER[62]. Wang et al. reported a potential-induced dynamic transformation of CoNi-MOFs into their (oxy)hydroxide counterparts, showing remarkable 5-hydroxymethylfurfural oxidation efficiency[67]. This stands in stark contrast to the pristine MOF frameworks, in which saturated, and ligand-blocked metal sites often fail to interact efficiently with reactant molecules. It is important to note, however, that the potential-induced structural evolution is often complex, occurring through multi-step pathways and involving various intermediate phases. This raises a critical question: *How can we best harness the structural evolution of MOFs to fully capitalize on its many advantages?*

Currently, many studies tend to designate the newly formed species or defect-sites as the real active centers, aiming to optimize the catalytic performance of MOFs. However, due to the lack of effective regulation and mitigation over the evolution pathway, MOF frameworks are often uncontrollably structural evolution, resulting in final phases that are not necessarily the most active evolved MOF structures. For example, Zhang et al. found that the intermediate phase α-$Co(OH)_2$ in-situ formed during MOF structural evolution exhibited the highest catalytic activity, rather than the subsequent β-$Co(OH)_2$ phase or the final CoOOH phase (Fig. 2d)[6]. If all metal–ligand coordination bonds and the corresponding framework collapse during/after electrocatalysis, the system is essentially no different from using the MOF as a precursor/template catalysis. In such cases, the intrinsic features of the MOF, such as porosity and high surface area, are not utilized at all. This represents a serious waste of the MOF structural potential and, similarly, does not align with the pathway toward the industrial application of MOFs as electrocatalysts.

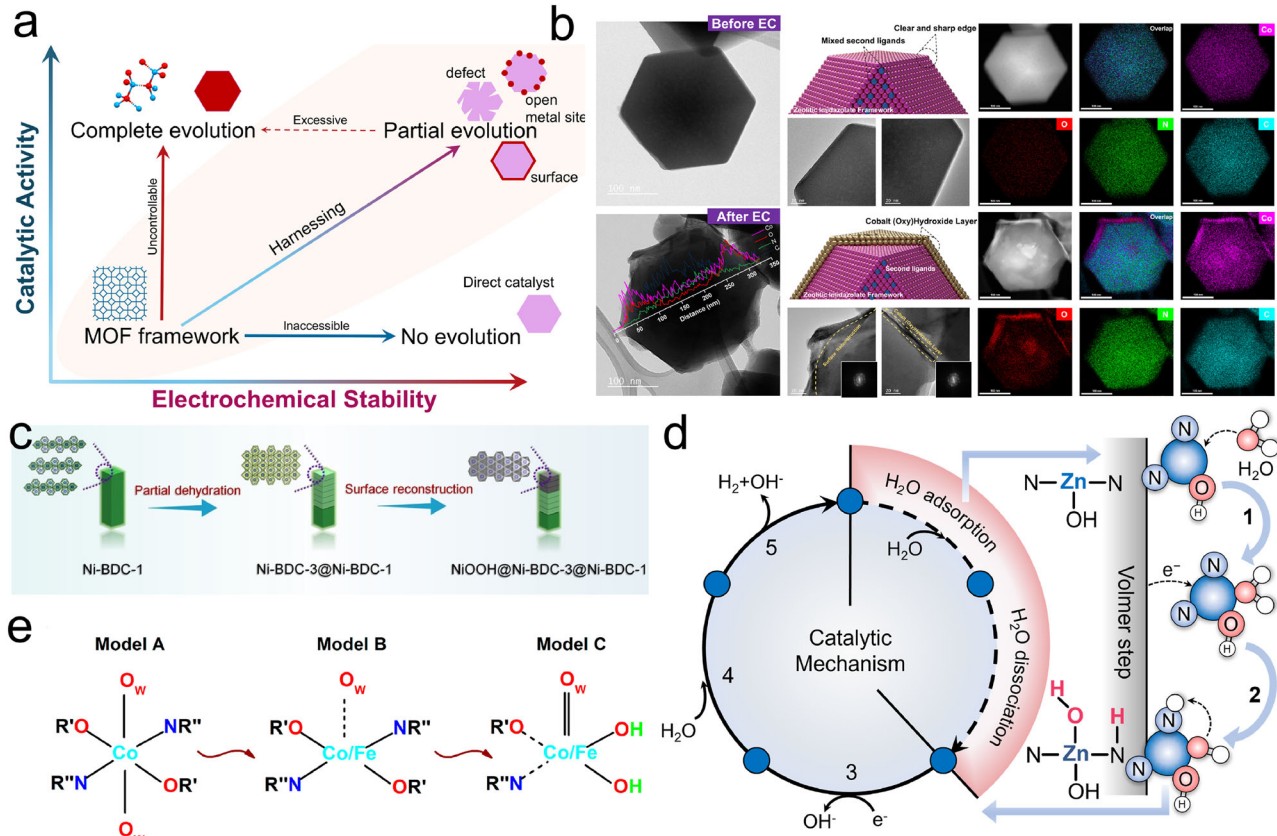

**Fig. 3 | Structural evolution pathways and their activation mechanisms.**
**a** Schematic illustration of the structural evolution pathways for MOF frameworks.
**b** Morphology evolution of mixed-ligand ZIF before and after 12 h amperometry[26].
Copyright 2024 Springer Nature. **c** Schematic illustration of the self-reconstruction
of Ni-BDC heterojunction[55]. Copyright 2022 Wiley-VCH GmbH. **d** Electrocatalytic
HER mechanism of open metal site ZIF in alkaline aqueous solution[82]. Copyright
2024 Wiley-VCH GmbH. **e** Schematic illustration of Fe assistant for in-situ elec-
trochemical activation[83]. Copyright 2019 American Chemical Society.

Therefore, this Perspective posits that the ideal future of MOF struc-
tural evolution lies in achieving full controllability, enabling the in-situ
generation of the desired/optimal evolved MOF structures as the catalyti-
cally active species, while avoiding unnecessary loss of its intrinsic advan-
tages (Fig. 3a). Such partial/surface structural evolution has been widely
reported in other metal-based electrocatalysts with systematic design stra-
tegies and well-established mechanistic insights across numerous
Reviews[68–71], along with demonstrated activity advantages. For example,
Chen et al. summarized the benefits of surface reconstruction in transition
metal-based catalysts and the advanced techniques used to identify the
resulting active species for the OER[68]. Similarly, Huang et al. emphasized
that such electrochemical self-adaptive of surface structures plays a key role
in achieving higher catalytic efficiency[69]. The same concept is applicable to
MOF systems, where ideally, a harnessing structural evolution would
requires sacrificing only a portion of the MOF (particularly surface regions
and select metal sites) to form a stable active species@MOF material. Such
systems can synergistically combine the catalytic advantages of the evolved
species (e.g., high-valent metal oxides or hydroxides) with the inherent
features of MOFs (e.g., high surface area and pore architecture), while also
enhancing electron transfer efficiency at their interfacial regions.

For example, in our previous study on mixed-ligand ZIF-67, we con-
fined the electrooxidation-induced structural evolution to the particle sur-
face, where a surface layer of cobalt (oxy)hydroxide was formed in-situ
during operation, without the complete reconstruction into nanosheets as
observed in pristine ZIF-67 (Fig. 3b)[26]. Similarly, Zhang et al. reported that
Ni-BDC-1 underwent self-reconstruction during OER, forming a shielding
NiOOH coating on the MOF surface; this MOF-based heterojunction
exhibited remarkable durability and OER activity (Fig. 3c)[55]. Such surface
electro-oxidation behavior is commonly observed in Ni-, Co-, and Fe-based

MOFs and is often referred to in the literature as "surface reconstruction";
however, it is essentially a form of structural evolution[72–74].

Under electrocatalytic conditions, the presence and engineering of
defects in MOFs play a decisive role in determining both the rate of struc-
tural evolution and the nature of the resulting active sites. In general, the
defects in MOFs are defined as "sites that locally break the regular periodic
arrangement of atoms or ions of the static crystalline parent framework due
to missing or displaced atoms or ions[75]." Structurally, MOF lattice defects are
typically categorized as missing-ligand (ML) defects and missing-cluster
(MC) defects[76]. ML defects occur when an organic ligand is removed,
leaving behind open/unsaturated metal sites (OMS) and corresponding
coordination vacancies on adjacent metal clusters. MC defects, on the other
hand, arise when a secondary building unit (SBU) or metal cluster, together
with its entire coordinating ligands, is removed, thereby generating one or
more OMSs. In essence, ML defects can evolve into MC defects, and both
types may coexist, with their distribution depending on the critical defect
concentration and their spatial arrangement, without a clear boundary
between them[77]. For instance, in our previous work we demonstrated that
thermal removal of a secondary ligand in mixed-ligand ZIF-8 engineers ML
and/or MC defects of different sizes, which can be deliberately tuned and
expanded by adjusting secondary ligand content and temperature[78].

Similar to traditional solid-state materials, MOF defects can be clas-
sified according to their size and dimensionality (point, line, planar, or
micro-/mesoscale volume defects), or by location (surface vs. internal)[75]. For
electrocatalysis, surface defects are particularly crucial, as they more readily
allow penetration and interaction with electrolyte species. Surface ML/MC
defects, either deliberately introduced during synthesis or generated in-situ
under operating conditions, can lower the activation energy for bond
rearrangements and create coordinatively unsaturated catalytic centers[79].

Huxley et al. discussed that such defects in MOFs can serve not only as intrinsic active sites but also as nucleation centers, accelerating the transformation of MOF materials into catalytically favorable phases[80]. In particular, defect-rich regions facilitate the coordination of carboxylate/hydroxyl species or introduce oxygen vacancies, both of which modulate the local electronic structure and improve charge transfer kinetics[76]. Of course, the presence of defects in MOFs is not always beneficial for catalysis, such as Pablo et al. confirmed that ML defects in COK-47-Ti act as sites for the rate-limiting charge recombination, and their elimination can improve HER activity[81].

Building on this, structural evolution to deliberate modified of the metal coordination environment, often in concert with defect engineering and OMS evolution, can further boost the MOF electrocatalytic performance. For example, we recently reported a defect-engineered ZIF featuring open $Zn-N_2$ sites that remain stable in aqueous electrolytes[82]. Upon applying a potential, these sites coordinate with $OH^-$ from the electrolyte, in-situ forming high-valent $HO-Zn-N_2$ species (Fig. 3d). These species retain their nature as unsaturated metal sites, making them favorable for water adsorption and dissociation during the hydrogen evolution reaction (HER), as confirmed by density functional theory simulations (DFT). Similarly, Zou et al. doped Fe into Co-MOF to regulate the bond strength between the central metal and coordinated $H_2O$[83]. The electrochemically activated CoFe-MOF-OH exhibited improved intrinsic OER activity due to the defect assisted in-situ formation of active metal hydroxide sites (Fig. 3e). In another case, Wang et al. observed that the uncoordinated moiety of carboxylate groups in Fe/Zn-MOF promoted hydroxyl activation and dissociation during OER, thereby accelerating the proton- and electron-transfer steps in electrocatalysis[84].

The aforementioned examples primarily involve partial structural evolution and defect-sites evolution, which is strategically regulated by external and internal strategies to steer the evolution pathway (such strategies will be discussed in detail later). At present, the major challenge is that too few MOF reports exhibit harnessing structural evolution, making it difficult to establish scalable systems or derive evolutionary mechanisms. Most studies still focus on using MOFs merely as precursors/templates for efficient electrocatalysts, often neglecting the intrinsic structural features of the MOF itself. Furthermore, given that industrial alkaline water electrolysis typically involves 20–40 wt% KOH solution[85], the strong interactions between $OH^-$ ions and metal centers make MOFs prone to structural degradation. However, it is precisely this current situation that underscores the urgency and significance of pursuing beneficial and harnessing structural evolution within MOFs. This Perspective argues that such an approach represents a promising path forward toward the long-term goal of scalable, MOF-based electrocatalysis in industrial applications.

## Operando/in-situ electrochemical characterization techniques

To better leverage the favorable structural evolution of structurally unstable MOFs, operando or in-situ characterization techniques coupled with electrochemical setups are essential for real-time monitoring and analysis (Table 1). Operando/in-situ electrochemical characterization refers to the direct observation of MOF surfaces/interfaces under simulated reaction processes or conditions, allowing for the identification of newly formed active sites, new phases, or reaction intermediates[86]. This technique is essential for elucidating the mechanisms of structural evolution and can significantly advance our understanding of MOF-based catalysis. In contrast, conventional ex-situ characterization techniques alone are insufficient and should only be considered complementary. Among the most informative and widely used techniques for probing structural evolution are operando/in-situ electrochemical Raman and synchrotron X-ray absorption fine spectroscopy (XAS/XAFS).

Raman spectroscopy, which relies on the inelastic scattering of monochromatic light to probe molecular vibrations, offers a powerful means to monitor structural and chemical changes in electrocatalysts under realistic conditions. In an operando configuration, the spectrometer is

**Table 1 | Operando/in-situ electrochemical techniques for MOF electrocatalysts**

| Target Information | Technique | Operando/In-Situ Capability | Lab-Based or Synchrotron | Strengths | Limitations |
|---|---|---|---|---|---|
| Metal oxidation states and coordination information[105] | XAS (XANES/EXAFS^a) | Yes (widely used) | Lab & Synchrotron | Element-specific, quantitative; ideal for monitoring metal sites; operando cells available | Requires synchrotron for high resolution; MOF model-dependent data fitting |
| Crystalline structure and phase change[106,107] | XRD, PDF^b | Limited operando setups | Lab & Synchrotron | Tracks long-range structural evolution; ideal for bulk structural changes | Poor sensitivity to local or amorphous features |
| Cluster/Nanoparticle formation[95,108] | TEM/HR-TEM/ED^c | Rare in-situ; mostly ex-situ | Lab & Advanced Centers | High spatial resolution; ideal for visualizing nanoscale evolution | In-situ liquid-cell TEM is complex and beam-sensitive for MOF; interpretation nontrivial |
| Functional group, M–L bond changes and reaction intermediates[82,109] | Raman/SERS, FTIR/SEIRAS^d | Yes (widely used) | Lab | Real-time tracking of ligand dissociation/adsorption; Sensitive to electrochemical adsorbates; useful for mechanistic insights | Interference in Raman; limited detection of symmetric/weakly IR-active species; Requires enhanced surface signal; overlaps common |
| Optoelectronic structure[26] | UV-vis, PL, UPS^e | Some in-situ studies | Lab | Tracks changes in optical absorption and electronic structure; fast and simple | Cannot directly provide structural information; PL sensitive to defects and quenching |
| Conductivity and Electron transport[110,111] | EIS^f, UV-vis | Yes | Lab | Evaluates ionic/electronic conductivity and interface behavior | Indirect structural info; needs complementary techniques |

^aXANES/EXAFS: X-ray Absorption Near Edge Structure/Extended X-ray Absorption Fine Structure.
^bPDF: Pair Distribution Function.
^cED: Electron Diffraction.
^dSEIRAS: Surface-Enhanced Infrared Absorption Spectroscopy.
^eUPS: Ultraviolet Photoelectron Spectroscopy.
^fEIS: Electrochemical Impedance Spectroscopy.
Summary of commonly used techniques to probe MOF structural and electronic evolution under reaction conditions, including the type of information obtained, applicability in operando/in-situ setups, and key strengths and limitations.

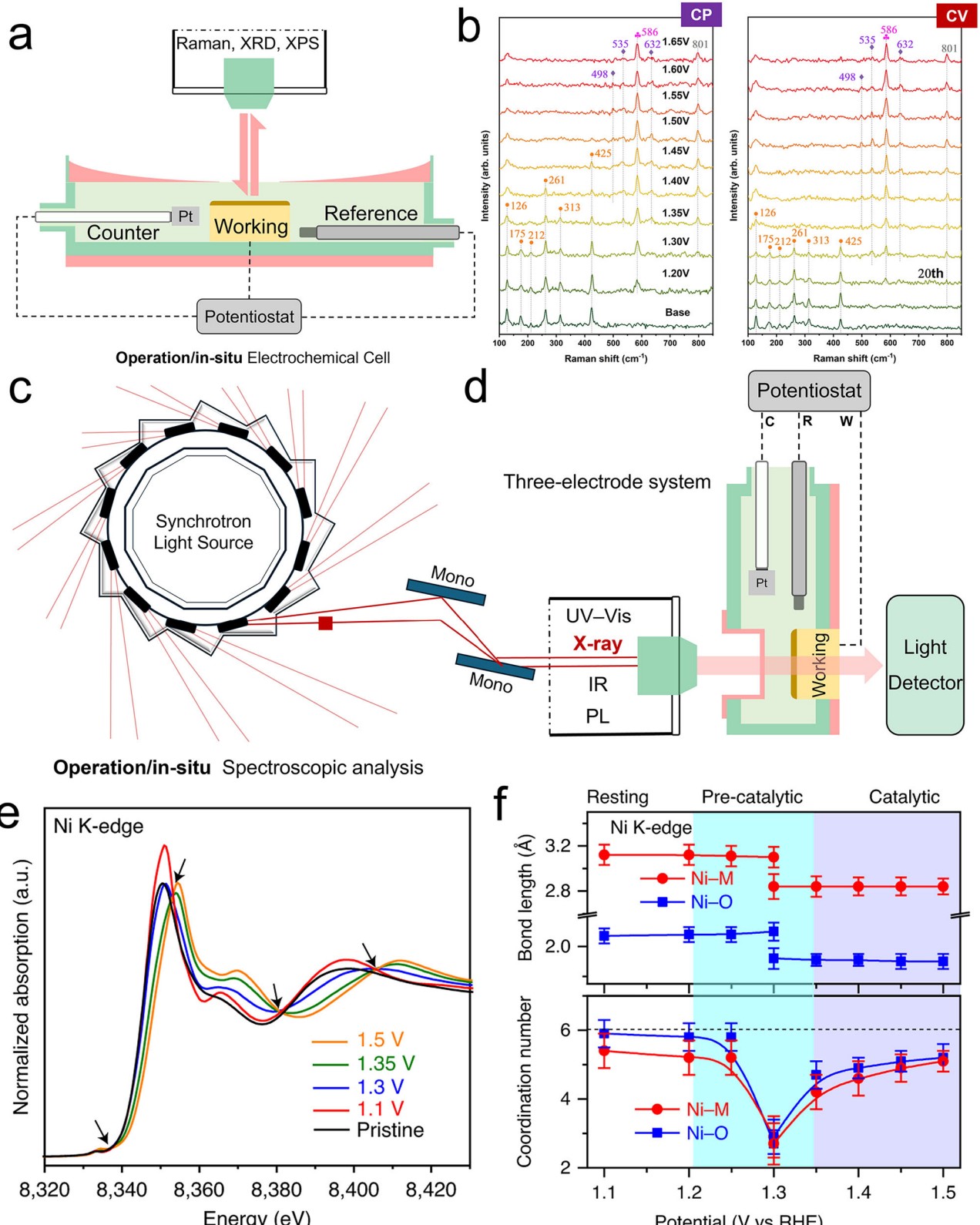

**Fig. 4 | Operando/in-situ spectroscopic techniques for probing MOF structural evolution. a** Operation/in-situ electrochemical spectroscopic cell designed for surface-sensitive techniques with a three-electrode system. **b** In-situ electrochemical Raman spectroscopy of ZIF-67 at various applied potentials from 1.20–1.65 V vs. RHE and 100 CV cycles at 0.85–1.55 V vs. RHE[26]. Copyright 2024 Springer Nature. **c** Synchrotron-accelerated X-ray source and **d** operation/in-situ electrochemical spectroscopic cell designed for transmission-based techniques with a three-electrode system. **e** Operando Ni K-edge XAS spectra at different applied potentials, and **f** the fitted changes in bond lengths and coordination numbers of the Ni–O/Ni–M coordination shells[74]. Copyright 2020 Springer Nature.

coupled with a specially designed electrochemical cell (Fig. 4a), enabling real-time tracking of intermediate species and structural motifs as the reaction proceeds under applied potential. Raman spectroscopy has become widely popular due to their flexibility and relatively low cost. Compared to infrared (IR) spectroscopy, Raman signals are insensitive to water, making the technique especially suitable for in-situ measurements in aqueous or electrolyte environments. As a result, it is especially useful for identifying surface species during the reactions, which provide distinct signals corresponding to changes in oxidation state and surface chemistry, such as the in-situ transformation of metal (oxy)hydroxides from $Co^{2+}$ to $Co^{3+}$ and $Co^{4+}$ (Fig. 4b)[26,87,88]. Moreover, Raman spectroscopy can probe vibrational modes of the MOF ligands, allowing for an indirect assessment of framework degradation. However, one limitation is that Raman intensity is influenced by various factors such as focusing conditions, sample surface roughness, or gas bubbles generated during electrochemical reactions[16]. Additionally, it usually requires a relatively high metal content in the sample to produce strong signals from metal–ligand bonds, making it unsuitable for quantitative analysis. High-sensitivity surface-enhanced Raman spectroscopy (SERS) may help address this issue by amplifying Raman signals, enabling the detection of subtle changes in trace surface species[89].

X-ray absorption spectroscopy (XAS), in contrast, is an element-specific probe that measures the absorption of X-rays as their energy is tuned across the absorption edge of a target element, revealing both the local electronic states (X-ray absorption near edge structure; XANES) and geometric coordination environment (Extended X-ray absorption fine structure; EXAFS). A more advanced yet resource-intensive technique is operando synchrotron-based XAS (Fig. 4c, d) that utilizes high-intensity X-rays from a synchrotron source to detect the electronic structure of metal elements at very low concentrations. In an operando setting, synchrotron-based XAS technique allows atomic-scale tracking of changes in oxidation state, valence, coordination number, and bond length under working electrochemical conditions. This technique has been well established in studies of other electrocatalytic oxide thin films, such as a combination of surface-sensitive XAS and surface scattering techniques revealed a coupled ionic diffusion-driven amorphization pathway in $SrIrO_3$ during the OER[90], and tracked the evolution of valence and covalence states in $LaFeO_3/LaNiO_3$ under potential control[91]. These precedents highlight the powerful capability of XAS in resolving atomic-scale transformations, and provide valuable information on structural evolution within MOF framework[27]. For example, Zhao et al. used wavelet transform analysis of XAS data at various applied potentials to track changes in coordination information, revealing a two-phase structural evolution in MOF-74: $Ni_{0.5}Co_{0.5}(OH)_2$ and $Ni_{0.5}Co_{0.5}OOH_{0.75}$ (Fig. 4e, f)[74]. In addition, XAS is also well-suited for verifying and monitoring the hydroxylation of open metal sites within MOFs, which are often challenging to identify using conventional techniques[82]. Recently, laboratory-based XAS systems have attracted attention as a more accessible and "low-cost" alternative to synchrotron-based X-ray facility[92,93]. For example, Malzer et al. demonstrated that modern lab-based XAS setups can achieve resolution comparable to that of synchrotron systems, expanding the potential for conducting operando electrocatalytic MOF studies without relying on national facilities[94]. Note that, however, that XAS data provides ensemble-averaged coordination information, and can be subject to interpretation or manipulation bias. Therefore, it is strongly recommended to use XAS as a "final" characterization, after obtaining complementary data from other techniques to support accurate XAS fitting.

In-situ liquid-phase transmission electron microscopy (LP-TEM) offers another powerful technique, enabling direct visualization of morphological, crystallographic, and local atomic structural changes in the real space as a high-energy electron beam through the sample. For example, Yang et al. combined in-situ electrochemical liquid-phase STEM and XAS to reveal the structural evolution mechanism from $Cu@Cu_2O$ nanocubes to polycrystalline metallic Cu nanograins under $CO_2$ reduction reaction[95]. However, this technique requires highly specialized reaction cells and operating conditions[86,96]. Moreover, many MOFs are electron-beam-

sensitive materials, limiting the widespread application of LP-TEM for studying structural evolution, particularly for observing the atomic arrangements of newly formed phases[97]. In the future, integrating ultra-low-dose or cryo-TEM with spherical aberration correction and electrochemical setups may mark a new milestone in revealing MOF evolution mechanisms at the atomic level.

In addition to the three techniques discussed above, other operando/in-situ techniques such as X-ray diffraction (XRD), X-ray photoelectron spectroscopy (XPS), infrared (IR), ultraviolet-visible (UV-vis), photoluminescence (PL) spectroscopy and so on can also provide valuable, albeit more limited, real-time MOF structural information[98]. A detailed overview of most characterization techniques and their corresponding target information is provided in Table 1. Integrating multiple operando/in-situ techniques with fundamental ex-situ characterization analysis enables better technical complementarity.

For the operando/in-situ electrochemical techniques applied to MOFs, it is strongly recommended to follow a three-step protocol: ex-situ (powder form) → *OCP* (electrode under open-circuit potential) → in-situ (electrode under applied potentials). This protocol is highly effective in identifying whether the evolved MOF structures are induced by the chemical environment (i.e., the electrolyte) or electrochemical environment (i.e., the applied potential)[99]. Moreover, it is crucial to use electrodes that closely mimic actual reaction conditions while maintaining the same electrolyte environment, in order to minimize discrepancies and ensure the accuracy of in-situ measurement results. Due to the volume limitations of most in-situ electrochemical cells, thin film/carbon paper electrodes and 2D metal mesh electrodes are commonly used as MOF powder loads (Fig. 2b). The former is more suitable for surface-sensitive techniques (e.g., XRD, Raman, and XPS in reflection mode in Fig. 4a), whereas the latter, owing to its excellent optical/electron transparency, is better suited for transmission-based techniques (e.g., IR, UV-vis, PL, XAS, and TEM in Fig. 4d), especially when beam penetration through the sample is required. Moving forward, the simultaneous integration of multiple characterization techniques into a single in-situ electrochemical cell represents a promising direction[86], as it can effectively prevent discrepancies in the observed MOF evolution pathways that may arise from switching between different cells or electrochemical environments.

## Current strategies for harnessing structural evolution

As exemplified earlier, due to the inherent nature of MOF structures, most pristine MOFs tend to rapidly degrade and transform into one or more new phases under even moderate or harsh electrocatalytic conditions, often following two-step or multi-step evolution pathways. This implies that the final phase cannot simply be assumed to represent the most catalytically active species, such as intermediate phases sometimes exhibiting superior activity due to partial retention of the parent MOF frameworks. Therefore, to better investigate the in-situ generated active centers and evolutionary mechanisms of MOFs in electrocatalysis, additional strategies are required to regulate and mitigate the evolution process. This enables controlled and beneficial structural evolution toward post-activation MOF structures that preserve their intrinsic framework advantages.

Currently, both external and internal strategies are being explored to harness the structural evolution process during electrocatalysis. Externally, commonly used electrochemical activation methods include cyclic voltammetry (CV), constant potential (CP), and pulsed potential (PP). CV activation, which continuously cycles the potential, can dynamically form a EDL on the MOF surface and accelerate structural evolution[5]. The extent of unwanted redox reactions under operating conditions can be partially avoided by carefully selecting the potential window, scan rate, and number of cycles. Compared to CV, CP activation offers greater stability and enables a more controlled evolution process to obtain the desired MOF structure by avoiding the cation/anion-concentrated chemical environment caused by constant potential[26]. Xia et al. compared CV and CP treatments for iron sulfides and their evolved oxides using operando XAS and modeling (Fig. 5a)[100]. They found that CV and CP directed distinct evolution

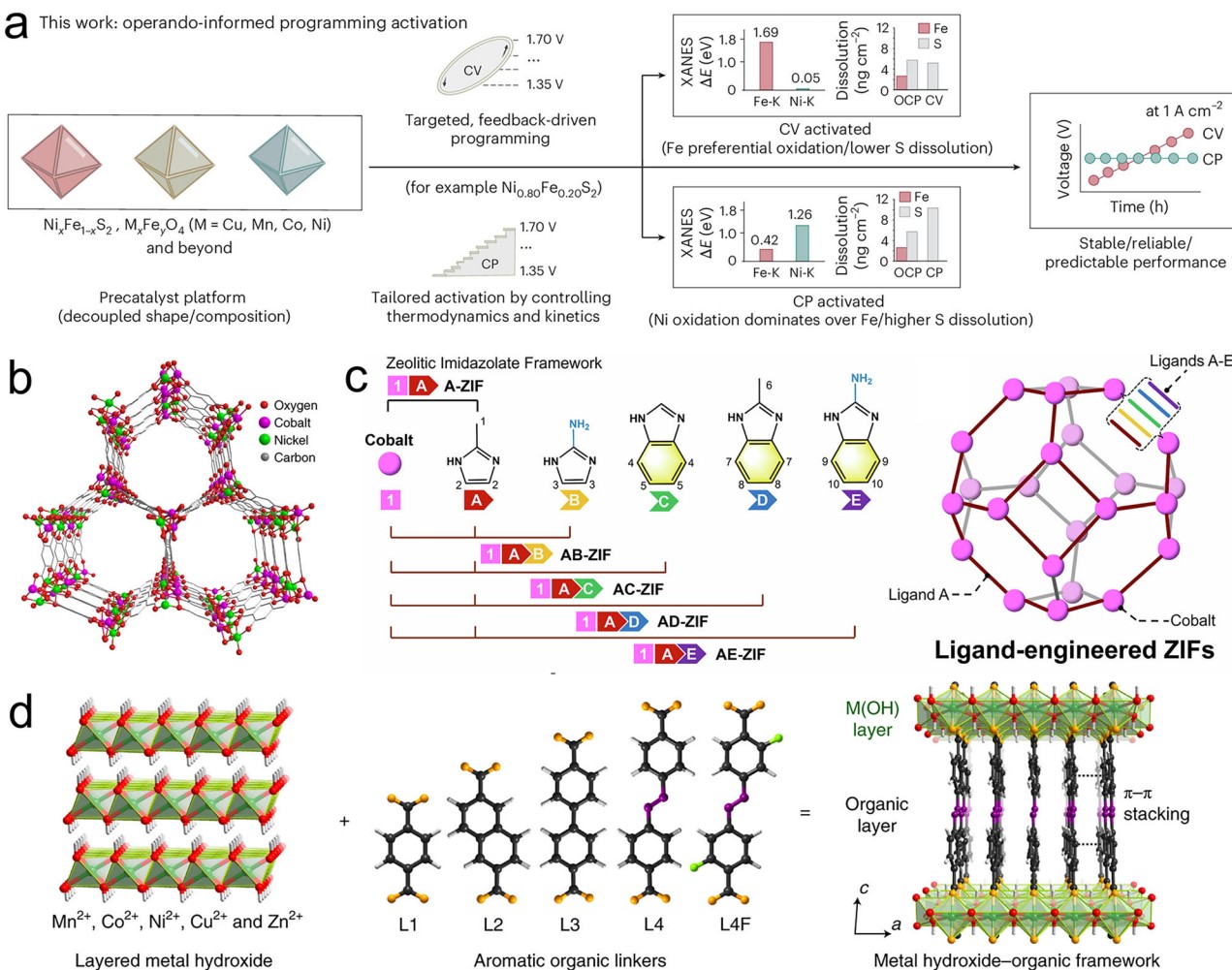

**Fig. 5 | Structural design and activation strategies. a** Activation strategies of catalysts via cyclic voltammetry (CV) and pulsed potential (PP)[100]. Copyright 2025 Springer Nature. **b** Crystal structure of $Ni_{0.5}Co_{0.5}$-MOF-74[74]. Copyright 2020 Springer Nature. **c** Schematic diagram of the various ligands A-E mixing and unit cell of each MOFs[26]. Copyright 2024 Springer Nature. **d** Schematic representation of the MOF assembly process, illustrating metal hydroxide layers comprising edge-sharing metal-octahedral chains crosslinked with neighboring chains via organic ligands[28]. Copyright 2020 Springer Nature.

pathways: CV promoted rapid surface Fe oxidation through repeated cycling, while CP maintained a steady-state environment that gradual activation, more controlled evolution with less Fe oxidation. In the MOF field, we previously reported on the structural evolution of ZIF-67 (Co) under CV and CP activation using in-situ Raman (Fig. 4b)[26]. Although no significant difference in the final evolved phase was observed (possibly due to limited sensitivity of the techniques and/or a narrow potential range), this does not mean the choice between CV and CP is interchangeable. On the contrary, more comparative studies across MOFs with different topologies under CV and CP treatment are needed in the future. Another method, PP activation, though rarely explored in MOF systems, has shown great promise in modulating reaction intermediates and kinetics in other electrocatalytic materials, as demonstrated by Casebolt et al. in the field of $CO_2$ reduction electrocatalysis[101,102], suggesting potential for future exploration in MOF-based systems. The choice of activation method can significantly affect the evolution pathway and ultimately the resulting MOF structure, which is particularly critical in electrocatalysis. Similarly, the chemical environment (e.g., electrolyte, pH, temperature, and pressure) can also yield similar results, as extensively discussed in previous Reviews[27,103].

Beyond these external factors, internal structural features of MOFs play an even more critical role in determining their structural robustness. And recent studies have primarily focused on structural modification strategies to enhance MOF stability and electrical conductivity, thereby facilitating more effective harnessing of structural evolution. For instance, in bimetallic MOFs, adjusting the Ni/Co ratio in NiCo-MOF-74 can direct the evolution pathway toward either hydroxide or oxyhydroxide phases, thereby tuning catalytic activity (Fig. 5b)[72,74]. Similarly, Binyamin et al. showed that tuning the Ni/Fe composition ratio in a 2D Zr-MOF directs electrochemical activation to form NiFeOOH, yielding a highly tunable pre-catalyst with optimal water oxidation activity[104]. Ligand engineering is a more general strategy: in our previous work, we embedded secondary ligands containing amine groups and π–π stacking aromatic rings into ZIF-67, which enhanced electrical conductivity and strengthened Co–N orbital hybridization (Fig. 5c)[26]. This mixed-ligand strategy mitigated electro-oxidation at high OER potentials and confined the structural evolution to the particle surface. Yuan et al. combined bimetallic Ni/Fe nodes with aromatic carboxylate ligands capable of strong π–π interactions, achieving highly active and stable MOFs due to Ni hydroxide modulation and the optimized binding of oxygenated intermediates (Fig. 5d)[28]. Similarly, Ma et al. used extended carboxylate ligands in Ni-MOFs to mitigate the phase transition toward α/β-$Ni(OH)_2$ during OER[87] and HER[88]. By facilitating electron transfer through the formation of internal heterojunctions, Zhang et al.[55] and Bao et al.[73] successfully mitigated the structural evolution of Ni-BDC, confining it to the MOF surface and enabling the formation of a protective NiOOH layer.

When these strategies are combined with operando/in-situ techniques, the evolved MOF structures can be effectively identified, and their true catalytic species can be evaluated through the correlation between current density and applied potential. This establishes a practical framework for

harnessing structural evolution and validating underlying mechanisms, applicable to both intermediate and final phases. Through the external (e.g., potential tuning) and internal (e.g., enhanced stability) strategies, it facilitates us obtaining high-active evolved MOF structures that retain the inherent framework, thereby unlocking their full potential in electrocatalytic applications.

## Summary and perspective

In summary, this Perspective highlights that the structural "instability" of MOFs under chemical and electrochemical environments should not be seen solely as a drawback, but rather as an opportunity to unlock their true catalytic potential. By harnessing and deliberately guiding structural evolution, MOFs can undergo beneficial transformations that enhance both catalytic performance and operational stability. Such evolution can result in the formation of new active sites, better exposure of under-coordinated/open metal centers, or even the emergence of composite structures such as core-shell architectures. These changes also modulate the geometric and electronic structure of the metal nodes, improving conductivity, redox behavior, and reactant binding, which are all critical to catalytic efficiency.

The progression of this evolution is governed by a combination of factors, including the intrinsic coordination chemistry and stability of the MOF, the chemical and electrochemical nature of the reaction environment, external conditions such as temperature, pressure, and gas atmosphere, as well as the characteristics of the applied potential (whether constant, step-wise, or gradually ramped). Therefore, capturing these complex, dynamic, and often subtle changes requires not only a strategic integration of ex-situ characterization but also the use of advanced operando/in-situ techniques. These approaches are essential to reveal both transient and stable species with high spatial and temporal resolution, enabling insight into the kinetics, thermodynamics, and transformation pathways of the newly formed sites, phases, or intermediates during the electrocatalytic process.

Crucially, we advocate a shift in the community's mindset: the pursuit of a perfectly stable MOF structure may obscure the actual source of catalytic activity. Instead, we can view MOFs as pre-catalysts that are intentionally designed to undergo controlled structural evolution, thereby forming active configurations under operating conditions. This dynamic nature, when harnessed appropriately, becomes a distinct advantage over more rigid catalytic materials. Recognizing the catalytic value embedded in this structural evolution not only strengthens the role of MOFs in electro-catalysis but also expands their potential across diverse applications such as photocatalysis and energy storage.

Looking ahead, the integration of structural evolution studies with predictive modeling and techno-economic analysis could significantly accelerate the development of scalable MOF-based catalysts. Computational modeling, such as DFT, machine learning, and kinetic Monte Carlo simulations can help predict dynamic changes in metal–ligand coordination, identify favorable reaction intermediates, and correlate atomic-scale transformations with macroscopic catalytic behavior. Although such methods are still relatively underexplored in the MOF community, they have proven powerful in other catalyst systems and hold great promise for developing scalable and cost-effective MOF-based electrocatalysts.

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

## Acknowledgements
Z.H. was financially supported by China Scholarship Council (Nos. 202106770017). This research was funded in part by the Austrian Science Fund (FWF: I5413-N and Cluster of Excellence MECS: 10.55776/COE5).

## Author contributions
Z.H. contributed wrote the initial manuscript. D.E. contributed the reviewing & editing the manuscript.

## Competing interests
The authors declare no competing interests.
