## [Transparent Peer Review file · Communications Chemistry]

Harnessing the structural evolution of metal–organic frameworks under electrocatalytic conditions

Corresponding Author: Professor Dominik Eder

Version 0:

Reviewer comments:

Reviewer #1

(Remarks to the Author)

Zheao Huang and Dominik Eder's perspective reframes the instability of Metal-Organic Frameworks (MOFs) in electrochemical process not as a drawback, but as a opportunity to create highly active catalytic species, such as metal (oxy)hydroxides, surface defects and more. They explain this structural evolution through rational design and activation strategies. The authors highlight in-situ techniques as crucial tools for real-time monitoring of structural evolution mechanisms, integrating these insights into a powerful asset for catalytic functionality. The paper proposing that the ideal future lies in achieving full controllability to generate optimal evolved MOF structures as active catalytic species, while preserving the intrinsic advantages of MOFs like high surface area and pore architecture.

1. Preservation of MOF Structure During Electrochemical Conversion

I suggest the authors expand their discussion on the preservation of framework integrity during electrochemical activation, particularly in the context of robust MOFs such as Zr-based frameworks. These MOFs, which can host redox-active metal species, offer a unique platform for achieving partial structural evolution such as surface transformations while retaining the overall crystallographic architecture. Clarifying how such systems can be leveraged to balance activity and structural fidelity would strengthen the perspective.

2. Layer Thickness and Structural Preservation

It would be valuable if the authors could comment on the influence of MOF film thickness on the outcome of structural evolution. Specifically, can the authors suggest an optimal thickness range that allows for effective electrochemical conversion (e.g., formation of active sites) while minimizing the collapse of the MOF framework and maintaining electrode stability?

3. Additional Example of Bimetallic Tuning

In line 269, the authors discuss the role of bimetallic systems in directing MOF structural evolution. A relevant and recent example that could enhance this discussion is:

Binyamin, Shahar, et al. "Nickel–Iron-Modified 2D Metal–Organic Framework as a Tunable Precatalyst for Electrochemical Water Oxidation." *ACS Applied Materials & Interfaces* 16.11 (2024): 13849–13857.

This study exemplifies how bimetallic interactions can modulate activation pathways and enhance catalytic performance, making it highly relevant to the concepts discussed.

overall I find this paper highly relevant and valuable to the MOF research community. It offers a timely and insightful perspective on the role of structural evolution in MOF-based materials under electrochemical conditions. Importantly, it challenges the conventional emphasis on stability by presenting a compelling case for intentionally harnessing structural transformation as a design principle. This approach is particularly impactful for guiding the development of MOF-derived materials, which are not always viewed through a unified lens. The paper contributes meaningfully to reshaping how we think about MOF functionality in electrocatalysis and related applications.

Reviewer #2

(Remarks to the Author)

In this review, two authors tried to illustrate the idea of using metal-organic frameworks as the precursor to generate active sites under electro-catalytic conditions, along with the characterization tools for such structural evolution. It is an interesting summary, despite a lack of a deeper insight into physical chemistry (e.g., defects evolution) and characterization tools. A major revision is suggested. The authors need to address the following concerns before any resubmission.

1) The title and the abstract claimed the "electrochemical conditions" which is very misleading. This paper only covers electrocatalytic reactions. It is worth noting that batteries and supercapacitors also involve electrochemical conditions. It is

recommended that the author can make a more precise title and a revised abstract, with a focus on electrocatalytic environments. All the discussions about electrochemical conditions in the main text should be revised accordingly.

2) On Fig. 2, what does author mean for “pre-catalysis” vs. “direct catalysts”. There appears to be typos. Furthermore, Fig. 2 covers both chemical treatments (e.g., pyrolysis, sulfurization) and electrocatalytic operations. However, the figure caption says, “Potential influencing factors of MOF evolution pathways during electrocatalysis and 208 their classification as electrocatalysts.”. This is contradictory and needs a revision.

3) On the characterization section, authors need to discuss some of the relevant summaries on the transformation of working catalysts (e.g., ACS Catalysis, 2022, 12 (13), 8007-8018; Nature Reviews Chemistry, 2021, 5256–276) and further then elucidate what is common feature and unique parts between pre-catalysts of MOF vs. oxides etc. The discussion of these characterization tools is at a very preliminary level, with no discussion on how it works even at the highest level. No discussion on the “in-situ” and “operando”. It is suggested that the author either cite and discuss these references or provides a more detailed discussion about these techniques (e.g., XAS) in Fig. 4. BTW, more citations are needed to support the claims in Table 1. It is a review paper, not an original research paper, even the latter needs proper citation.

4) Fundamental insights hold the key to establishing predictable models and generalized strategies. Recently, using oxide thin films as model systems (Science Advances 2021, 7 (2), eabc7323; J. Phys. Chem. C 2024, 128, 13, 5515–5523; ACS Appl. Mater. Interfaces 2024, 16, 16, 21273–21282), studies have been able to uncover the transformation at the atomic level. It is recommended authors discuss these previous studies and further envision the how to create model systems to study MOF transformation as well as the characterization challenges to obtain a molecular-level picture.

5) The defect can play a key role in altering how fast the MOF undergoes the change and the as formed active sites. However, the introduction of defect chemistry is bare in this review. It is suggested that the author should discuss the role of defects in MOF evolution under electrocatalytic conditions. It is recommended that authors read these following papers (e.g., Chem. Mater. 2014, 26, 1, 348–360; Accounts of Chemical Research, 2021, 54 (15), 3039-3049; Commun Mater 5, 247 (2024).) and discuss the role of defects in structural changes of MOF under electrocatalytic conditions in greater detail.

6) Last but not least, what is the difference between “pyrolysis” and “carbonization” in the main text and also in Fig. 2? What is the “catalytic black box”? This paper needs careful proofreading before resubmission. It contains lots of undefined concepts and the utilization of misleading terminology.

Reviewer #3

(Remarks to the Author)

This Perspective discusses the structural “instability” of metal–organic frameworks (MOFs) under electrochemical conditions and reframes it as a controllable “structural evolution” process that can be leveraged to improve catalytic performance. Key factors influencing MOF stability—such as electrode stability, and chemical and electrochemical environments—are examined, alongside operando/in-situ characterization techniques and both external and internal strategies to regulate the evolution process. The manuscript argues that partial or surface-limited evolution can combine the advantages of newly formed active species with the intrinsic properties of MOFs, offering a pathway toward more efficient and stable electrocatalysts. The topic is timely, and the manuscript provides a broad overview of relevant literature. However, the discussion on “harnessing” structural evolution is not sufficiently developed or supported with systematic evidence. The manuscript would benefit from a clearer conceptual framework, more critical analysis, and stronger emphasis on unique contributions after minor revision.

1. Can the authors explicitly compare this Perspective with recent publications on MOF structural evolution and reconstruction, and clearly indicate what new insights or conceptual frameworks this work offers?
2. Under what specific conditions do MOFs become unstable, and what are the underlying causes? Could the authors clarify how partial reconstruction, complete reconstruction, and fully stable structures influence catalytic performance, and under what circumstances MOFs function best as electrocatalysts?
3. Could the authors provide a more detailed analysis of how MOF structural evolution behaves under realistic industrial conditions and outline possible engineering strategies for controlling these processes?
4. Can the discussion of operando/in-situ techniques go beyond listing methods, by analyzing their applicability to MOFs, identifying technical limitations, and recommending suitable combinations for different stages of structural evolution?
5. It would be better if the authors expand the literature coverage by citing important and closely related works, for example: J. Am. Chem. Soc. 2023, 145, 37, 20624–20633; Nat. Commun. 2019, 10, 5048; Energy Environ. Sci. 2022, 15, 3830–3841; Adv. Mater. 2021, 33, 2007344.

Version 1:

Reviewer comments:

Reviewer #1

(Remarks to the Author)

I truly appreciate the hard work the researchers put into this study. I believe it is a valuable and reliable review that deserves

to be published as it stands.

Reviewer #2

(Remarks to the Author)

The authors have tried to address the prior comments; however, these are not complete. A revision is suggested.

I suggest the authors take a close look into the suggested reference and make corresponding changes.

1) On page 12, please discuss the relevant defect types in MOF structures, instead of a blurry statement saying that defects are important.

2) on page 15, "such as operando XAS revealed a coupled ionic diffusion-driven amorphization pathway in SrIrO₃ during the OER". After a careful read of the paper, it is not the case. This should be "a combination of surface-sensitive XAS and surface scattering techniques" instead of "operando XAS". This needs a correction.

3) The author should include a section to summarize and discuss the stability challenges of MOF as electrocatalysts in acid and alkaline electrolytes.

Reviewer #3

(Remarks to the Author)

In this resubmitted version, the authors have explained the peer review comments accordingly and improved the manuscript. The quality of this article has been improved significantly and is suitable for publication. So I recommend the direct acceptance of this manuscript.

Version 2:

Reviewer comments:

Reviewer #2

(Remarks to the Author)

The authors have made proper revisions. This paper is recommended for publication.

Point-by-point response to Reviewers' comments

Reviewers' comments:

Reviewer #1

General comment. Zheao Huang and Dominik Eder's perspective reframes the instability of Metal-Organic Frameworks (MOFs) in electrochemical process not as a drawback, but as an opportunity to create highly active catalytic species, such as metal (oxy)hydroxides, surface defects and more. They explain this structural evolution through rational design and activation strategies. The authors highlight in-situ techniques as crucial tools for real-time monitoring of structural evolution mechanisms, integrating these insights into a powerful asset for catalytic functionality. The paper proposing that the ideal future lies in achieving full controllability to generate optimal evolved MOF structures as active catalytic species, while preserving the intrinsic advantages of MOFs like high surface area and pore architecture.

Overall I find this paper highly relevant and valuable to the MOF research community. It offers a timely and insightful perspective on the role of structural evolution in MOF-based materials under electrochemical conditions. Importantly, it challenges the conventional emphasis on stability by presenting a compelling case for intentionally harnessing structural transformation as a design principle. This approach is particularly impactful for guiding the development of MOF-derived materials, which are not always viewed through a unified lens. The paper contributes meaningfully to reshaping how we think about MOF functionality in electrocatalysis and related applications.

Response: Thank you for your positive feedback on our work. We truly appreciate your recognition of our approach and its potential impact on the MOF research community. We have addressed the comments you provided below, and hope you are satisfied with the revisions and improvements meet your expectations.

1. Preservation of MOF Structure During Electrochemical Conversion

I suggest the authors expand their discussion on the preservation of framework integrity during electrochemical activation, particularly in the context of robust MOFs such as Zr-based frameworks. These MOFs, which can host redox-active metal species, offer a unique platform for achieving partial structural evolution such as surface transformations while retaining the overall crystallographic architecture. Clarifying how such systems can be leveraged to balance activity and structural fidelity would strengthen the perspective.

Response: Thank you for this valuable suggestion. We have expanded our discussion to include examples of Zr-based MOFs and their stability in electrochemical environments. We agree that the transformation of host or supported guest redox-metal species within Zr-based frameworks into catalytically active phases is a very interesting direction, since these robust hosts can maintain their long-range crystallinity while enabling partial evolution of the incorporated species. However, this topic lies somewhat outside the central scope of the present manuscript, which is focused on the structural evolution of MOF backbones and their intrinsic metal sites or framework structures under electrocatalysis. A comprehensive discussion of guest-to-active-species transformations would involve a wide variety of cases and thus diverge from the perspective we intend to provide here. In this work, we emphasize MOF-intrinsic structural evolution and surface reconstruction processes, such as the generation of high-valent hydroxides/oxyhydroxides, which have been predominantly reported for transition-metal-based frameworks (Ni, Co, Fe) that are highly favorable for electrocatalysis. These systems and their structural-activity relationships have already been described in detail in the present manuscript.

Page 3: Direct catalysts refer to MOFs that are used directly in reactions without prior treatments, with their structural integrity verified by comparing pre- and post-reaction characterizations, often supplemented by in-situ techniques. Examples include S-doped NiBDC nanosheets ¹ and missing-ligand layered-pillared CoBDC ². Zr-based MOFs, such as NU-1000, also represent an important class of direct catalysts, exhibiting great chemical stability in water under neutral and even acidic pH conditions ³⁻⁴. These studies aim to optimize the electronic structure of MOFs while maintaining their framework under specific conditions to achieve catalytic activity. However, such stability must be validated on a case-by-case basis for each MOF and reaction type, such as Zr-based MOFs fail in strong alkaline media, making them far less versatile than conventional metal oxide/sulfide catalysts ⁵.

2. Layer Thickness and Structural Preservation

It would be valuable if the authors could comment on the influence of MOF film thickness on the outcome of structural evolution. Specifically, can the authors suggest an optimal thickness range that allows for effective electrochemical conversion (e.g., formation of active sites) while minimizing the collapse of the MOF framework and maintaining electrode stability?

Response: We appreciate the reviewer’s insightful suggestion regarding MOF films and the influence of their thickness on structural evolution during electrochemical processes. However, our laboratory has not yet synthesized or worked with MOF thin films for electrocatalysis. Instead, our studies have primarily employed common electrode supports such as carbon paper and metal foams, where we have gained in-depth understanding of their behavior and mechanisms, which informed the “MOF Electrode Stability” section of this manuscript. As this work is a Perspective, we aimed to base our discussion on systems we have directly handled or applied, so as to provide an original viewpoint. While comprehensive Review on MOF films are already available in the literature ⁶, we currently lack the experimental background to discuss MOF films and their thickness effects in sufficient depth. This is certainly an interesting direction for future work, and we hope the reviewer understands our present scope and focus.

3. Additional Example of Bimetallic Tuning

In line 269, the authors discuss the role of bimetallic systems in directing MOF structural evolution. A relevant and recent example that could enhance this discussion is: Binyamin, Shahar, et al. "Nickel–Iron-Modified 2D Metal–Organic Framework as a Tunable Precatalyst for Electrochemical Water Oxidation." ACS Applied Materials & Interfaces 16.11 (2024): 13849–13857.

This study exemplifies how bimetallic interactions can modulate activation pathways and enhance catalytic performance, making it highly relevant to the concepts discussed.

Response: We appreciate the reviewer’s suggestion and fully agree that the referenced work is highly relevant to our discussion of bimetallic tuning and structural evolution phenomena in MOFs. Accordingly, we have incorporated a brief description of this study into the appropriate section of the manuscript and have cited it to strengthen the discussion.

Page 10: For instance, in bimetallic MOFs, adjusting the Ni/Co ratio in NiCo-MOF-74 can direct the evolution pathway toward either hydroxide or oxyhydroxide phases, thereby tuning catalytic activity (**Figure 5b**) ⁷⁻⁸. Similarly, Binyamin et al. showed that tuning the Ni/Fe composition ratio in a 2D Zr-MOF directs electrochemical activation to form NiFeOOH, yielding a highly tunable pre-catalyst with optimal water oxidation activity ⁹.

Reviewer #2

General comment. In this review, two authors tried to illustrate the idea of using metal-organic frameworks as the precursor to generate active sites under electro-catalytic conditions, along with the characterization tools for such structural evolution. It is an interesting summary, despite a lack of a deeper insight into physical chemistry (e.g., defects evolution) and characterization tools. A major revision is suggested. The authors need to address the following concerns before any resubmission.

Response: We sincerely appreciate your valuable comments and suggestions. We have addressed all the concerns you raised, with particular emphasis on the defect and characterization tools. These revisions have been crucial in improving the quality and clarity of our manuscript.

1. The title and the abstract claimed the “electrochemical conditions” which is very misleading. This paper only covers electrocatalytic reactions. It is worth noting that batteries and supercapacitors also involve electrochemical conditions. It is recommended that the author can make a more precise title and a revised abstract, with a focus on electrocatalytic environments. All the discussions about electrochemical conditions in the main text should be revised accordingly.

Response: We agree with the reviewer’s point that our previous use of the term “electrochemical conditions” was imprecise, as the manuscript does not cover batteries or supercapacitors. We have therefore revised the title, abstract, and all relevant parts of the main text to replace “electrochemical conditions” with “electrocatalytic conditions,” ensuring the terminology accurately reflects the scope of our work.

Title: *Harnessing Structural Evolution of MOFs under Electrocatalytic Conditions*

Page 2: In this Perspective, we reframe MOF evolution in electrocatalytic conditions as a controllable pathway for accessing highly active catalytic species, such as metal (oxy)hydroxides, surface defects, and open metal centers.

Page 10: MOF Structural Evolution under Electrocatalytic Conditions

Page 20: As exemplified earlier, due to the inherent nature of MOF structures, most pristine MOFs tend to rapidly degrade and transform into one or more new phases under even moderate or harsh electrocatalytic conditions, often following two-step or multi-step evolution pathways.

2. On Fig. 2, what does author mean for “pre-catalysis” vs. “direct catalysts”. There appears to be typos. Furthermore, Fig. 2 covers both chemical treatments (e.g., pyrolysis, sulfurization) and electrocatalytic operations. However, the figure caption says, “Potential influencing factors of MOF evolution pathways during electrocatalysis and 208 their classification as electrocatalysts.”. This is contradictory and needs a revision.

Response: We thank the reviewer for the careful examination of our manuscript and for pointing out this issue. In our work, “direct catalysts” refer to MOFs that participate in catalytic reactions without additional pre-treatments (particularly those that would disrupt the inherent framework topology), while fully maintaining their framework or structural stability over a certain period, without irreversible evolution. In this case, the intrinsic metal sites of the MOF directly function as the electrocatalytic active centers, rather than the evolved structures. By contrast, “pre-catalysis”, the main focus of this Perspective, describes MOFs with their framework intact that enter the catalytic reaction, but undergo partial or complete structural evolution during operation. This process is often accompanied by the formation of new metal species, defects, and/or unsaturated metal sites, which, whether acting alone or in concert with the remaining MOF framework, typically enhance electrocatalytic activity.

We agree that the current caption of **Figure 2a** (“Potential influencing factors of MOF evolution pathways during electrocatalysis and their classification as electrocatalysts”) is inaccurate, as the figure also includes chemical treatments in addition to electrocatalytic operations. We have revised it to: “Classification of MOFs as electrocatalysts, associated treatment methods, and potential factors influencing MOF evolution pathways.” This better reflects the content shown in **Figure 2a**.

Page 10: Figure 2: (a) Classification of MOFs as electrocatalysts, associated treatment methods, and potential factors influencing MOF evolution pathways.

3. On the characterization section, authors need to discuss some of the relevant summaries on the transformation of working catalysts (e.g., ACS Catalysis, 2022, 12 (13), 8007-8018; Nature Reviews Chemistry, 2021, 5256–276) and further then elucidate what is common feature and unique parts between pre-catalysts of MOF vs. oxides etc. The discussion of these characterization tools is at a very preliminary level, with no discussion on how it works even at the highest level. No discussion on the “in-situ” and “operando”. It is suggested that the author either cite and discuss these references or provides a more detailed discussion about these techniques (e.g., XAS) in Fig. 4. BTW, more citations are needed

to support the claims in Table 1. It is a review paper, not an original research paper, even the latter needs proper citation.

Response: We sincerely appreciate the reviewer's valuable comments on the characterization section and for recommending relevant literature. We have cited the suggested references and briefly discussed the differences between MOF pre-catalysts and oxide-based pre-catalysts. While we acknowledge the reviewer's concern that our discussion on characterization was not sufficiently comprehensive and lacked coverage of "in-situ" and "operando" techniques, we would like to clarify that, as this work is a Perspective rather than a comprehensive Review, our intention was not to provide an exhaustive description of each characterization method. Instead, the purpose of the "Characterization" section is to highlight representative in-situ techniques relevant to MOF structural evolution, outline their functions and limitations, and point out MOF-specific considerations.

The detailed working principles and experimental implementation of such techniques, including their applications in electrocatalysis and specifically in MOFs, have already been extensively discussed in existing Review articles¹⁰⁻¹¹. Repeating these would dilute the other focal points of our Perspective and make the manuscript unnecessarily lengthy, which we wish to avoid.

Nevertheless, in response to the reviewer's suggestions, we have expanded our descriptions of the two primary techniques emphasized in our manuscript, XAS and Raman spectroscopy, and added discussion regarding "operando" measurements.

Furthermore, we have revised **Table 1**, incorporated additional relevant references to support the entries, and ensured proper citation throughout the section.

Page 4: While the concept of pre-catalyst and associated terms are not unique to MOFs and is also well established for transition-metal oxides, hydroxides, and chalcogenides, both can undergo electrochemically induced structural evolution to generate catalytically active species¹²⁻¹³. The key distinction lies in the presence of organic ligands in MOFs, which not only provide tunable coordination and porosity but also introduce additional degradation pathways such as ligand detachment or framework collapse that are absent in purely inorganic oxides. Moreover, the weaker coordination bonds in MOFs can facilitate more pronounced atomic rearrangements or phase transformations under operating conditions, whereas oxide pre-catalysts often retain part of their crystalline lattice as a structural backbone¹⁴.

Page 15: Raman spectroscopy, which relies on the inelastic scattering of monochromatic light to probe molecular vibrations, offers a powerful means to monitor structural and

chemical changes in electrocatalysts under realistic conditions. In an operando configuration, the spectrometer is coupled with a specially designed electrochemical cell (**Figure 4a**), enabling real-time tracking of intermediate species and structural motifs as the reaction proceeds under applied potential. Raman spectroscopy has become widely popular due to their flexibility and relatively low cost. Compared to infrared (IR) spectroscopy, Raman signals are insensitive to water, making the technique especially suitable for in-situ measurements in aqueous or electrolyte environments

X-ray absorption spectroscopy (XAS), in contrast, is an element-specific probe that measures the absorption of X-rays as their energy is tuned across the absorption edge of a target element, revealing both the local electronic states (X-ray absorption near edge structure; XANES) and geometric coordination environment (Extended X-ray absorption fine structure; EXAFS). A more advanced yet resource-intensive technique is operando synchrotron-based XAS (**Figures 4c and 4d**) that utilizes high-intensity X-rays from a synchrotron source to detect the electronic structure of metal elements at very low concentrations. In an operando setting, synchrotron-based XAS technique allows atomic-scale tracking of changes in oxidation state, valence, coordination number, and bond length under working electrochemical conditions.

Page 19: Table 1. Overview of operando/in-situ electrochemical techniques for MOF catalysis.

4. Fundamental insights hold the key to establishing predictable models and generalized strategies. Recently, using oxide thin films as model systems (*Science Advances* 2021, 7 (2), eabc7323; *J. Phys. Chem. C* 2024, 128, 13, 5515–5523; *ACS Appl. Mater. Interfaces* 2024, 16, 16, 21273–21282), studies have been able to uncover the transformation at the atomic level. It is recommended authors discuss these previous studies and further envision the how to create model systems to study MOF transformation as well as the characterization challenges to obtain a molecular-level picture.

Response: We thank the reviewer for providing these highly valuable references. Although these studies primarily focus on oxide thin films, we agree that their revealed transformation mechanisms and applied techniques are highly relevant to the theme of this Perspective, especially in the context of their operation as electrocatalysts. We have selected two examples, *Science Advances* 2021, 7 (2), eabc7323 and *J. Phys. Chem. C* 2024, 128 (13), 5515–5523, both of which investigate surface evolution and interfacial transformation during electrocatalysis using XAFS. These have been incorporated as an introductory part of our discussion on the importance of employing XAFS to study MOF structural evolution. We believe this is the most appropriate way to connect oxide thin film

system insights with our envisioned strategies for probing MOF transformations and addressing the associated characterization challenges.

Page 15: In an operando setting, synchrotron-based XAS technique allows atomic-scale tracking of changes in oxidation state, valence, coordination number, and bond length under working electrochemical conditions. This technique has been well established in studies of other electrocatalytic oxide thin films, such as operando XAS revealed a coupled ionic diffusion-driven amorphization pathway in SrIrO₃ during the OER¹⁵, and tracked the evolution of valence and covalence states in LaFeO₃/LaNiO₃ under potential control¹⁶. These precedents highlight the powerful capability of XAS in resolving atomic-scale transformations, and provide valuable information on structural evolution within MOF framework¹¹.

5. The defect can play a key role in altering how fast the MOF undergoes the change and the as formed active sites. However, the introduction of defect chemistry is bare in this review. It is suggested that the author should discuss the role of defects in MOF evolution under electrocatalytic conditions. It is recommended that authors read these following papers (e.g., Chem. Mater. 2014, 26, 1, 348–360; Accounts of Chemical Research, 2021, 54 (15), 3039-3049; Commun Mater 5, 247 (2024).) and discuss the role of defects in structural changes of MOF under electrocatalytic conditions in greater detail.

Response: We sincerely thank you for highlighting the importance of discussing defect chemistry in the context of MOF structural evolution under electrocatalytic conditions and for recommending these relevant publications. In response, we have added a new discussion before the open metal sites MOF, where we describe the MOF defects, nature of the formed active sites and their influence on the kinetics of MOF structural evolution. We believe this addition makes our work more comprehensive. The papers you suggested are highly relevant to the topic, especially *Commun. Mater.* 5, 247 (2024), which provides an in-depth discussion of the benefits of MOF defects. We have now cited all three recommended papers in the newly added defect-related discussion.

Page 12: Under electrocatalytic conditions, the presence and engineering of defects in MOFs play a decisive role in determining both the rate of structural evolution and the nature of the resulting active sites. Defects such as missing ligands/clusters and ligand vacancy sites, often introduced intentionally during synthesis or generated in-situ during operation, can lower the activation energy for bond rearrangements, promote the penetration of electrolyte species, and create coordinatively unsaturated metal centers¹⁷⁻¹⁸.

Huxley et al. discussed that such defects in MOFs can serve not only as intrinsic active sites but also as nucleation centers, accelerating the transformation of MOF materials into catalytically favorable phases ¹⁹. In particular, defect-rich regions facilitate the coordination of carboxylate/hydroxyl species or introduce oxygen vacancies, both of which modulate the local electronic structure and improve charge transfer kinetics ²⁰.

Building on this, deliberate modification of the metal coordination environment, often in concert with defect engineering, can further boost catalytic performance. For example, we recently reported a defect-engineered ZIF featuring open Zn–N₂ sites that remain stable in aqueous electrolytes. ²¹ Upon applying a potential, these sites coordinate with OH[−] from the electrolyte, in-situ forming high-valent HO–Zn–N₂ species (**Figure 3d**). These species retain their nature as unsaturated metal sites, making them favorable for water adsorption and dissociation during the hydrogen evolution reaction (HER), as confirmed by density functional theory simulations (DFT). Similarly, Zou et al. doped Fe into Co-MOF to regulate the bond strength between the central metal and coordinated H₂O ²². The electrochemically activated CoFe-MOF-OH exhibited improved intrinsic OER activity due to the defect assisted in-situ formation of active metal hydroxide sites (**Figure 3e**). In another case, Zhang et al. observed that the uncoordinated moiety of carboxylate groups in Fe/Zn-MOF promoted hydroxyl activation and dissociation during OER, thereby accelerating the proton- and electron-transfer steps in electrocatalysis ²³.

6. Last but not least, what is the difference between “pyrolysis” and “carbonization” in the main text and also in Fig. 2? What is the ““catalytic black box””? This paper needs careful proofreading before resubmission. It contains lots of undefined concepts and the utilization of misleading terminology.

Response: We sincerely thank the reviewer for raising questions and providing valuable reminders regarding terminology definitions in our manuscript. Although “pyrolysis” and “carbonization” in MOF research both involve thermal treatment, their outcomes differ. “Carbonization” generally refers to high-temperature treatment (typically above 700 °C) that removes the inherent metal atoms from the MOF and leaves only a nitrogen and/or carbon framework, such as N-doped carbon. While “pyrolysis” is a broader concept that can include carbonization, it often refers to partial ligand removal or structural oxidation within the MOF framework during thermal treatment. However, we agree that these processes partially overlap. To avoid confusion, we have removed the term “carbonization” from both the main text and **Figure 2a**, incorporating it under the description of pyrolysis for a clearer presentation.

Regarding “catalytic black box,” we originally intended it to describe the uncertainty and difficulty of probing fragile MOF structures during catalysis. We acknowledge that this expression can be ambiguous and potentially misleading, so we have replaced it with a clearer, more precise description in the text. In addition, we have carefully proofread the manuscript to eliminate other undefined concepts and misleading terminology, ensuring greater clarity and accuracy.

Page 3: Precursor catalysts refer to MOFs that are deliberately converted, e.g. via pyrolysis, sulfurization, phosphidation, or other treatments, into inorganic composites comprising small metal or metal oxide species dispersed within a carbonaceous matrix, prior to catalysis²⁴⁻²⁶.

Figure 2: (a) Classification of MOFs as electrocatalysts, associated treatment methods, and potential factors influencing MOF evolution pathways. (b) Selection and preparation of MOF working electrodes in a typical three-electrode system. (c) 100 CV cycles of the ZIF-67 between 0.925 and 1.525 V vs. RHE²⁷. (d) Precatalytic evolution of ZIF-67 to α - and β -Co(OH)₂ and their further oxidation and OER activity²⁷. Copyright 2019 American Chemical Society.

Page 14: To better leverage the favorable structural evolution of structurally unstable MOFs, operando or in-situ characterization techniques coupled with electrochemical setups are essential for real-time monitoring and analysis (Table 1).

Reviewer #3

General comment. This Perspective discusses the structural “instability” of metal–organic frameworks (MOFs) under electrochemical conditions and reframes it as a controllable “structural evolution” process that can be leveraged to improve catalytic performance. Key factors influencing MOF stability—such as electrode stability, and chemical and electrochemical environments—are examined, alongside operando/in-situ characterization techniques and both external and internal strategies to regulate the evolution process. The manuscript argues that partial or surface-limited evolution can combine the advantages of newly formed active species with the intrinsic properties of MOFs, offering a pathway toward more efficient and stable electrocatalysts. The topic is timely, and the manuscript provides a broad overview of relevant literature. However, the discussion on “harnessing” structural evolution is not sufficiently developed or supported with systematic evidence. The manuscript would benefit from a clearer conceptual framework, more critical analysis, and stronger emphasis on unique contributions after minor revision.

Response: We sincerely thank the reviewer for comments on our manuscript. We believe that, based on our detailed responses provided below, the reviewer will gain a clearer understanding of our discussion on “harnessing” structural evolution and see that we have provided sufficient evidence to support this concept.

1. Can the authors explicitly compare this Perspective with recent publications on MOF structural evolution and reconstruction, and clearly indicate what new insights or conceptual frameworks this work offers?

Response: We thank the reviewer for this important question. Among recent publications closely related to our Perspective, one notable example is *Insight into the surface-reconstruction of metal–organic framework-based nanomaterials for the electrocatalytic oxygen evolution reaction* (June 1, 2023)¹¹. While this Review is valuable, it is somewhat dated and focuses mainly on surface reconstruction, which is a specific form of structural evolution. Our manuscript incorporates the most recent examples of MOF structural evolution and provides a broader, more comprehensive discussion that goes beyond surface reconstruction. Furthermore, this 2023 Review is primarily a review summarizing prior studies, with limited opinion-driven discussion. In contrast, our Perspective draws extensively on our own experimental work and related research publications^{21, 28}, enabling us to address overlooked but crucial aspects such as MOF electrode preparation,

environmental stability, and “operational” details of in-situ characterization, all of which are critical for understanding and leveraging structural evolution.

Another relevant recent publication is *Evolving metal–organic frameworks for highly active oxygen evolution* (May 7, 2025)²⁹. This work, while also a summary publication, is largely confined to the OER reaction. Our manuscript, in comparison, extends the discussion to both OER, HER and other reactions, highlighting not only the generation of high-valent hydroxy/oxide species under electro-oxidation but also the evolution of unsaturated metal sites through strong interactions under potential. These active sites are particularly relevant to HER activity and mechanisms.

Most importantly, our Perspective consolidates recent findings including our own to propose explicit strategies for harnessing and controlling structural evolution in MOFs to achieve optimal and targeted active structures for electrocatalysis. As this work is a Perspective, we link observation, mechanistic understanding, and regulation, which has not been systematically addressed in the recent summary Reviews.

2. Under what specific conditions do MOFs become unstable, and what are the underlying causes? Could the authors clarify how partial reconstruction, complete reconstruction, and fully stable structures influence catalytic performance, and under what circumstances MOFs function best as electrocatalysts?

Response: We thank the reviewer for this question. These points are all addressed in our manuscript, but we are happy to provide a consolidated explanation here. MOFs generally become unstable and/or degradation when exposed to harsh electrochemical environments. The chemical environment refers to factors such as the electrolyte composition, solvent, pH, coexisting ions, and reaction products. The electrochemical environment further involves the application of potential, which can induce dynamic structural and chemical changes. Both acts as external factors that affect MOF integrity under electrocatalytic conditions. Chemically, instability often arises from metal leaching, ligand exchange, or protonation/deprotonation of coordination bonds. For example, soft metal centers may undergo hydrolysis or ligand displacement in alkaline media, while certain ions in the electrolyte can competitively coordinate to the metal nodes. Electrochemically, applying a potential lead to the formation of an EDL at the electrode–electrolyte interface. This local microenvironment can differ markedly from the bulk electrolyte, concentrating reactive ions (e.g., OH⁻) at the MOF surface, shifting local pH, and promoting nucleophilic attack on metal–ligand bonds. This can drive partial or complete ligand dissociation, ion exchange, or redox transformation of metal centers into (oxy)hydroxides, oxides, or other active species.

Regarding catalytic performance, the impact of partial reconstruction, complete reconstruction, and full structural stability is linked to how the active metal sites and their coordination structures evolve: 1) Partial reconstruction can adjust the electronic structures of surface metal sites while retaining most of the original MOF topology, porosity, and high surface area. The evolved active sites that can thus combine the intrinsic advantages of MOFs with the high activity/conductivity of the newly coordinated species. 2) complete reconstruction typically leads to the loss of MOF structural features, producing active sites and catalysts similar in behavior to conventional metal (oxy)hydroxides or oxides. 3) Fully stable structures can preserve the original framework entirely, which is an ideal structure in principle. However, in many cases, saturated ligand-coordinated metal sites as active centers do not achieve optimal activity compared with traditional materials.

Our Perspective argues that the most promising pathway for MOF-based electrocatalysis lies in partial and harnessing structural evolution. This can couple the catalytic advantages of high-valent (oxy)hydroxides, isolated single atoms, or unsaturated metal sites with the structural benefits of MOFs. This coordinated approach can maximize both activity and stability, offering a rational strategy for designing next-generation MOF electrocatalysts.

3. Could the authors provide a more detailed analysis of how MOF structural evolution behaves under realistic industrial conditions and outline possible engineering strategies for controlling these processes?

Response: We thank the reviewer for highlighting the importance of realistic industrial conditions, which is indeed crucial for advancing MOFs toward large-scale applications. At present, MOF industrial-/commercialization has seen notable progress mainly in gas adsorption and separation, whereas their industrial use in electrocatalysis remains challenging. Our work is based on laboratory-scale studies, and we acknowledge the current lack of examples under true industrial conditions. Nevertheless, we envision that engineering strategies for controlling MOF structural evolution at scale could include designing robust frameworks with tailored metal-ligand chemistry, employing conductive and corrosion-resistant supports, integrating protective coatings to modulate local reaction environments, and optimizing reactor configurations. While these concepts are speculative, they provide a possible direction for future research bridging laboratory insights with industrial feasibility.

4. Can the discussion of operando/in-situ techniques go beyond listing methods, by analyzing their applicability to MOFs, identifying technical limitations, and recommending suitable combinations for different stages of structural evolution?

Response: We appreciate the reviewer's interest in state-of-the-art characterization techniques for probing MOF structural evolution during electrocatalysis. At present, the most widely used and effective methods for such studies are operando X-ray absorption and Raman spectroscopy¹¹, whose advantages, limitations, and applicability have been discussed in detail in the manuscript. For a broader overview, we have summarized other commonly employed techniques in **Table 1**. As the reviewer notes, there are indeed advanced emerging approaches, such as ultra-wideline NMR under static and magic-angle spinning conditions at low temperatures and fast repetition rates³⁰, which can sensitively resolve the coordination environment around metal or single-atom sites and thus be highly valuable for monitoring phenomena such as hydroxylation of open metal sites in MOFs during catalysis. Another example is valence-to-core X-ray emission spectroscopy (vtc-XES), which provides enhanced sensitivity for distinguishing C, N, and O coordination in metal centers in materials, offering higher precision than XAS when specific light-element coordination needs to be resolved³¹. However, these techniques are still rarely applied to MOFs, remain relatively specialized, and are not yet broadly accessible, including in our own studies. From our Perspective, we have therefore chosen not to devote extensive discussion to such less-established techniques, except where they may be essential for detecting specific intermediates or configurations. For most MOF systems, we recommend using in-situ/operando Raman, IR, and XRD to determine structural and compositional changes, supplemented by operando XAS fitting and electron microscopy imaging, to investigate the mechanism of MOF structural evolution under electrocatalytic conditions.

5. It would be better if the authors expand the literature coverage by citing important and closely related works, for example: *J. Am. Chem. Soc.* 2023, 145, 37, 20624–20633; *Nat. Commun.* 2019, 10, 5048; *Energy Environ. Sci.* 2022, 15, 3830–3841; *Adv. Mater.* 2021, 33, 2007344.

Response: We appreciate the reviewer's suggestions and have carefully considered each of the recommended works. The paper *J. Am. Chem. Soc.* 2023, 145, 37, 20624–20633 reports a potential-induced reversible dynamic structural transformation of oxalate-based CoNi-MOFs into hydroxides/oxyhydroxides, which aligns closely with the theme of our manuscript. Likewise, *Energy Environ. Sci.* 2022, 15, 3830–3841 describes the structural evolution of FeZn bimetallic MOFs in electrocatalysis, and *Adv. Mater.* 2021, 33, 2007344 summarizes the behavior of various pre-catalysts undergoing complete reconstruction. These three works have been incorporated into our references to broaden the scope and relevance of the manuscript. As for *Nat. Commun.* 2019, 10, 5048, which reports a missing-

ligand Co-MOF exhibiting stable OER activity, we have already cited it in the current manuscript as an example of a direct catalyst (see **Page 3**, missing-ligand layered-pillared CoBDC).

Page 4: Pre-catalysts refer to MOFs that undergo structural changes, ranging from subtle to substantial, during exposure to an electrochemical environment and/or throughout the electrocatalytic reaction. In the literature, a variety of terms have been used to describe this process such structural evolution, structural reconstruction, structural transformation, and structural degradation ³².

Page 11: Wang et al. reported a potential-induced dynamic transformation of CoNi-MOFs into their (oxy)hydroxide counterparts, showing remarkable 5-hydroxymethylfurfural oxidation efficiency ³³. This stands in stark contrast to the pristine MOF frameworks, in which saturated, and ligand-blocked metal sites often fail to interact efficiently with reactant molecules.

Page 13: In another case, Wang et al. observed that the uncoordinated moiety of carboxylate groups in Fe/Zn-MOF promoted hydroxyl activation and dissociation during the OER, thereby accelerating the proton- and electron-transfer steps occurring in electrocatalysis ²³.

References

1. Cheng, F.; Peng, X.; Hu, L.; Yang, B.; Li, Z.; Dong, C.-L.; Chen, J.-L.; Hsu, L.-C.; Lei, L.; Zheng, Q.; Qiu, M.; Dai, L.; Hou, Y., Accelerated water activation and stabilized metal-organic framework via constructing triangular active-regions for ampere-level current density hydrogen production. *Nat. Commun.* **2022**, *13* (1), 6486.
2. Xue, Z.; Liu, K.; Liu, Q.; Li, Y.; Li, M.; Su, C.-Y.; Ogiwara, N.; Kobayashi, H.; Kitagawa, H.; Liu, M.; Li, G., Missing-linker metal-organic frameworks for oxygen evolution reaction. *Nat. Commun.* **2019**, *10* (1), 5048.
3. Aparna, R. K.; Karmakar, A.; Arsha, R. T.; Kundu, S.; Mandal, S., Copper nanoparticle-embellished Zr-based metal-organic framework for electrocatalytic hydrogen evolution reaction. *Chem. Commun.* **2023**, *59* (69), 10444-10447.
4. Tsai, M.-D.; Wu, K.-C.; Kung, C.-W., Zirconium-based metal-organic frameworks and their roles in electrocatalysis. *Chem. Commun.* **2024**, *60* (64), 8360-8374.
5. Hayes, D.; Alia, S.; Pivovar, B.; Richards, R., Targeted synthesis, characterization, and electrochemical analysis of transition-metal-oxide catalysts for the oxygen evolution reaction. *Chem Catalysis* **2024**, *4* (2), 100905.
6. Li, W.; Mukherjee, S.; Ren, B.; Cao, R.; Fischer, R. A., Open Framework Material Based Thin Films: Electrochemical Catalysis and State-of-the-art Technologies. *Advanced Energy Materials* **2022**, *12* (4), 2003499.
7. Zhao, S.; Tan, C.; He, C.-T.; An, P.; Xie, F.; Jiang, S.; Zhu, Y.; Wu, K.-H.; Zhang, B.; Li, H.; Zhang, J.; Chen, Y.; Liu, S.; Dong, J.; Tang, Z., Structural transformation of highly active metal-organic framework electrocatalysts during the oxygen evolution reaction. *Nature Energy* **2020**, *5* (11), 881-890.
8. Chen, T.; Xu, H.; Li, S.; Zhang, J.; Tan, Z.; Chen, L.; Chen, Y.; Huang, Z.; Pang, H., Tailoring the Electrochemical Responses of MOF-74 Via Dual-Defect Engineering for Superior Energy Storage. *Adv. Mater.* **2024**, *36* (31), 2402234.
9. Binyamin, S.; Shimoni, R.; Liberman, I.; Ifraemov, R.; Tashakory, A.; Hod, I., Nickel-Iron-Modified 2D Metal-Organic Framework as a Tunable Precatalyst for Electrochemical Water Oxidation. *ACS Appl. Mater. Interfaces* **2024**, *16* (11), 13849-13857.
10. Song, J.; Qian, Z.-X.; Yang, J.; Lin, X.-M.; Xu, Q.; Li, J.-F., In situ/Operando Investigation for Heterogeneous Electro-Catalysts: From Model Catalysts to State-of-the-Art Catalysts. *ACS Energy Letters* **2024**, *9* (9), 4414-4440.
11. Liu, Y.; Wang, S.; Li, Z.; Chu, H.; Zhou, W., Insight into the surface-reconstruction of metal-organic framework-based nanomaterials for the electrocatalytic oxygen evolution reaction. *Coord. Chem. Rev.* **2023**, *484*, 215117.
12. Liu, L.; Corma, A., Structural transformations of solid electrocatalysts and photocatalysts. *Nature Reviews Chemistry* **2021**, *5* (4), 256-276.
13. Wan, G.; Zhang, G.; Chen, J. Z.; Toney, M. F.; Miller, J. T.; Tassone, C. J., Reaction-Mediated Transformation of Working Catalysts. *ACS Catalysis* **2022**, *12* (13), 8007-8018.
14. Zheng, W.; Lee, L. Y. S., Metal-Organic Frameworks for Electrocatalysis: Catalyst or Precatalyst? *ACS Energy Letters* **2021**, *6* (8), 2838-2843.
15. Wan, G.; Freeland, J. W.; Kloppenburg, J.; Petretto, G.; Nelson, J. N.; Kuo, D.-Y.; Sun, C.-J.; Wen, J.; Diulus, J. T.; Herman, G. S.; Dong, Y.; Kou, R.; Sun, J.; Chen, S.; Shen, K. M.; Schlom, D. G.; Rignanese, G.-M.; Hautier, G.; Fong, D. D.; Feng, Z.; Zhou, H.; Suntivich, J., Amorphization mechanism of SrIrO₃ electrocatalyst: How oxygen redox initiates ionic diffusion and structural reorganization. *Science Advances* **7** (2), eabc7323.

16. Che, Q.; van den Bosch, I. C. G.; Le, P. T. P.; Lazemi, M.; van der Minne, E.; Birkhölzer, Y. A.; Nunnenkamp, M.; Peerlings, M. L. J.; Safonova, O. V.; Nachtegaal, M.; Koster, G.; Baeumer, C.; de Jongh, P.; de Groot, F. M. F., In Situ X-ray Absorption Spectroscopy of LaFeO₃ and LaFeO₃/LaNiO₃ Thin Films in the Electrocatalytic Oxygen Evolution Reaction. *J. Phys. Chem. C* **2024**, *128* (13), 5515-5523.
17. Wan, G.; Sun, C.-J.; Freeland, J. W.; Fong, D. D., Defect-Driven Oxide Transformations and the Electrochemical Interphase. *Acc. Chem. Res.* **2021**, *54* (15), 3039-3049.
18. Maier, J., Pushing Nanoionics to the Limits: Charge Carrier Chemistry in Extremely Small Systems. *Chem. Mater.* **2014**, *26* (1), 348-360.
19. Portillo-Vélez, N. S.; Obeso, J. L.; de los Reyes, J. A.; Peralta, R. A.; Ibarra, I. A.; Huxley, M. T., Benefits and complexity of defects in metal-organic frameworks. *Communications Materials* **2024**, *5* (1), 247.
20. Qiu, X.; Wang, R., From construction strategies to applications: Multifunctional defective metal-organic frameworks. *Coord. Chem. Rev.* **2025**, *526*, 216356.
21. Huang, Z.; Wang, Z.; Zhou, Q.; Rabl, H.; Naghdi, S.; Yang, Z.; Eder, D., Engineering of HO–Zn–N₂ Active Sites in Zeolitic Imidazolate Frameworks for Enhanced (Photo)Electrocatalytic Hydrogen Evolution. *Angew. Chem. Int. Ed.* **2025**, *64* (7), e202419913.
22. Zou, Z.; Wang, T.; Zhao, X.; Jiang, W.-J.; Pan, H.; Gao, D.; Xu, C., Expediting in-Situ Electrochemical Activation of Two-Dimensional Metal–Organic Frameworks for Enhanced OER Intrinsic Activity by Iron Incorporation. *ACS Catalysis* **2019**, *9* (8), 7356-7364.
23. Wang, Y.; Zhao, L.; Ma, J.; Zhang, J., Confined interface transformation of metal–organic frameworks for highly efficient oxygen evolution reactions. *Energy Environ. Sci.* **2022**, *15* (9), 3830-3841.
24. Song, Z.; Zhang, L.; Doyle-Davis, K.; Fu, X.; Luo, J.-L.; Sun, X., Recent Advances in MOF-Derived Single Atom Catalysts for Electrochemical Applications. *Advanced Energy Materials* **2020**, *10* (38), 2001561.
25. Huang, H.; Shen, K.; Chen, F.; Li, Y., Metal–Organic Frameworks as a Good Platform for the Fabrication of Single-Atom Catalysts. *ACS Catalysis* **2020**, *10* (12), 6579-6586.
26. Huang, Z.; Zhou, Q.; Wang, J.; Yu, Y., Fermi-level-tuned MOF-derived N-ZnO@NC for photocatalysis: A key role of pyridine-N-Zn bond. *J. Mater. Sci. Technol.* **2022**, *112*, 68-76.
27. Zheng, W.; Liu, M.; Lee, L. Y. S., Electrochemical Instability of Metal–Organic Frameworks: In Situ Spectroelectrochemical Investigation of the Real Active Sites. *ACS Catalysis* **2020**, *10* (1), 81-92.
28. Huang, Z.; Wang, Z.; Rabl, H.; Naghdi, S.; Zhou, Q.; Schwarz, S.; Apaydin, D. H.; Yu, Y.; Eder, D., Ligand engineering enhances (photo) electrocatalytic activity and stability of zeolitic imidazolate frameworks via in-situ surface reconstruction. *Nat. Commun.* **2024**, *15* (1), 9393.
29. Yang, P.; Yang, C.; Wu, Z.; Tang, Z., Evolving metal-organic frameworks for highly active oxygen evolution. *Matter* **2025**, *8* (5).
30. Koppe, J.; Yakimov, A. V.; Gioffrè, D.; Usteri, M.-E.; Vosegaard, T.; Pintacuda, G.; Lesage, A.; Pell, A. J.; Mitchell, S.; Pérez-Ramírez, J.; Copéret, C., Coordination

environments of Pt single-atom catalysts from NMR signatures. *Nature* **2025**, *642* (8068), 613-619.

31. Li, S.; Liu, T.; Zhang, W.; Wang, M.; Zhang, H.; Qin, C.; Zhang, L.; Chen, Y.; Jiang, S.; Liu, D.; Liu, X.; Wang, H.; Luo, Q.; Ding, T.; Yao, T., Highly efficient anion exchange membrane water electrolyzers via chromium-doped amorphous electrocatalysts. *Nat. Commun.* **2024**, *15* (1), 3416.

32. Liu, X.; Meng, J.; Zhu, J.; Huang, M.; Wen, B.; Guo, R.; Mai, L., Comprehensive Understandings into Complete Reconstruction of Precatalysts: Synthesis, Applications, and Characterizations. *Adv. Mater.* **2021**, *33* (32), 2007344.

33. Wang, Y.; Ma, J.; Cao, X.; Chen, S.; Dai, L.; Zhang, J., Bionic Mineralization toward Scalable MOF Films for Ampere-Level Biomass Upgrading. *J. Am. Chem. Soc.* **2023**, *145* (37), 20624-20633.

Point-by-point response to Reviewers' comments

Reviewers' comments:

Reviewer #1

General comment. I truly appreciate the hard work the researchers put into this study. I believe it is a valuable and reliable review that deserves to be published as it stands.

Response: We sincerely thank you for your positive feedback and encouraging comments on our work. We greatly appreciate your recognition of our efforts and are delighted that you find our perspective valuable and reliable.

Reviewer #2

General comment. The authors have tried to address the prior comments; however, these are not complete. A revision is suggested. I suggest the authors take a close look into the suggested reference and make corresponding changes.

Response: We sincerely apologize that our previous revision did not fully meet your expectations. At the same time, we truly appreciate your careful review of our revised manuscript and your constructive feedback, which has helped us to further improve the quality of the work. We have now carefully followed your new suggestions, revised the manuscript, and added new section and a more comprehensive summary. We hope that these revisions address your concerns and make the manuscript more satisfactory.

1. On page 12, please discuss the relevant defect types in MOF structures, instead of a blurry statement saying that defects are important.

Response: In the revised manuscript, we have rewritten and expanded the section on the types of defects in MOFs and included representative examples. These revisions are incorporated in the section *MOF Structural Evolution under Electrocatalytic Conditions*, and we hope this more comprehensive discussion addresses your concern.

Page 14: Under electrocatalytic conditions, the presence and engineering of defects in MOFs play a decisive role in determining both the rate of structural evolution and the nature of the resulting active sites. In general, the defects in MOFs are defined as “sites that locally break the regular periodic arrangement of atoms or ions of the static crystalline parent framework due to missing or displaced atoms or ions ¹.” Structurally, MOF lattice defects are typically categorized as missing-ligand (ML) defects and missing-cluster (MC) defects ². ML defects occur when an organic ligand is removed, leaving behind open/unsaturated metal sites (OMS) and corresponding coordination vacancies on adjacent metal clusters. MC defects, on the other hand, arise when a secondary building unit (SBU) or metal cluster, together with its entire coordinating ligands, is removed, thereby generating one or more OMSs. In essence, ML defects can evolve into MC defects, and both types may coexist, with their distribution depending on the critical defect concentration and their spatial arrangement, without a clear boundary between them ³. For instance, in our previous work we demonstrated that thermal removal of a secondary ligand

in mixed-ligand ZIF-8 engineers ML and/or MC defects of different sizes, which can be deliberately tuned and expanded by adjusting secondary ligand content and temperature ⁴.

Similar to solid-state materials, MOF defects can be classified according to their size and dimensionality (point, line, planar, or micro-/mesoscale volume defects), or by location (surface vs. internal) ¹. For electrocatalysis, surface defects are particularly crucial, as they more readily allow penetration and interaction with electrolyte species. Surface ML/MC defects, either deliberately introduced during synthesis or generated in-situ under operating conditions, can lower the activation energy for bond rearrangements and create coordinatively unsaturated catalytic centers ⁵. Huxley et al. discussed that such defects in MOFs can serve not only as intrinsic active sites but also as nucleation centers, accelerating the transformation of MOF materials into catalytically favorable phases ⁶. In particular, defect-rich regions facilitate the coordination of carboxylate/hydroxyl species or introduce oxygen vacancies, both of which modulate the local electronic structure and improve charge transfer kinetics ². Of course, the presence of defects in MOFs is not always beneficial for catalysis, such as Pablo et al. confirmed that ML defects in COK-47-Ti act as sites for the rate-limiting charge recombination, and their elimination can improve HER activity ⁷.

Building on this, deliberate modification of the metal coordination environment, often in concert with defect engineering and OMS evolution, can further boost the MOF electrocatalytic performance. For example, we recently reported a defect-engineered ZIF featuring open Zn-N₂ sites that remain stable in aqueous electrolytes. ⁸ Upon applying a potential, these sites coordinate with OH⁻ from the electrolyte, in-situ forming high-valent HO-Zn-N₂ species (**Figure 3d**). These species retain their nature as unsaturated metal sites, making them favorable for water adsorption and dissociation during the hydrogen evolution reaction (HER), as confirmed by density functional theory simulations (DFT). Similarly, Zou et al. doped Fe into Co-MOF to regulate the bond strength between the central metal and coordinated H₂O ⁹. The electrochemically activated CoFe-MOF-OH exhibited improved intrinsic OER activity due to the defect assisted in-situ formation of active metal hydroxide sites (**Figure 3e**). In another case, Wang et al. observed that the uncoordinated moiety of carboxylate groups in Fe/Zn-MOF promoted hydroxyl activation

and dissociation during OER, thereby accelerating the proton- and electron-transfer steps in electrocatalysis ¹⁰.

2. on page 15, "such as operando XAS revealed a coupled ionic diffusion-driven amorphization pathway in SrIrO₃ during the OER". After a careful read of the paper, it is not the case. This should be "a combination of surface-sensitive XAS and surface scattering techniques" instead of "operando XAS". This needs a correction.

Response: We are grateful for your careful reading and for pointing this out. Following your suggestion, we have corrected the statement in the manuscript to “a combination of surface-sensitive XAS and surface scattering techniques” instead of “operando XAS.” The revised description now accurately reflects the referenced work.

Page 17: This technique has been well established in studies of other electrocatalytic oxide thin films, such as a combination of surface-sensitive XAS and surface scattering techniques revealed a coupled ionic diffusion-driven amorphization pathway in SrIrO₃ during the OER ¹¹, and tracked the evolution of valence and covalence states in LaFeO₃/LaNiO₃ under potential control ¹².

3. The author should include a section to summarize and discuss the stability challenges of MOF as electrocatalysts in acid and alkaline electrolytes.

Response: We appreciate the reviewer’s valuable suggestion. We fully agree that a dedicated discussion of MOF stability in acid and alkaline electrolytes is important for our manuscript. Accordingly, we have added an additional section entitled “*MOF Stability in Electrolytes*”, where we summarize the stability examples and current challenges of MOFs as electrocatalysts under acidic and alkaline electrolyte conditions.

Page 6: MOF Stability in Electrolytes

In electrocatalysis, the electrolyte not only serves as an ionic conductor and reactant source but also strongly affects catalytic performance. To enhance activity, strong acidic or alkaline electrolytes are commonly employed as the aqueous solution, because they can provide high ionic conductivity due to the abundance of protons (H⁺) and hydroxide ions (OH⁻), acting as direct reactants to accelerate reaction kinetics ¹³. Therefore, before

evaluating electrocatalytic reactions, it is essential to assess not only the stability of MOFs in aqueous media but also their durability as catalysts under acidic and alkaline electrolytes.

Compared with neutral water, H^+ and OH^- exhibit much stronger destructive effects on MOFs by competing with ligands for coordination to metal centers, ultimately leading to framework degradation¹⁴⁻¹⁵. Strengthening metal–ligand bonds is thus a key strategy to improve MOF stability in electrolytes. Both the charge density of the metal ion and the hydrophobicity of the ligand significantly influence the robustness of coordination bonds¹⁴. For example, MIL-101 with Cr^{3+} –O coordination developed by Leus et al. exhibited excellent stability for over two months in aqueous solutions across pH 0–12¹⁶. Similar metal strategies have been applied to MOFs based on high-valence Zr^{4+} , Fe^{3+} , and Al^{3+} nodes¹⁷⁻¹⁹. On the ligand side, introducing hydrophobic groups can provide steric protection to metal sites. Zhong et al., for instance, developed pH-stable UiO-66 (Zr) variants by tuning the position of trifluoromethyl substituents²⁰. However, hydrophobic modification may also hinder the approach of electrolyte molecules to MOF catalytic sites, which, while enhancing stability, could slow catalytic reactions by limiting reactant accessibility. Post-synthetic modifications have also proven effective: Liu et al. transformed PCN-426 (Mg) into robust Fe- and Cr-MOFs using postsynthetic metathesis and oxidation strategies, achieving stability in both strong acids and bases due to the formation of inert Fe^{3+} –O and Cr^{3+} –O bonds²¹.

Currently, although numerous reports claim pH-stable MOFs, most rely solely on powder X-Ray diffraction (PXRD) patterns as evidence. More rigorous validation should involve porosity analyses (e.g., N_2 physisorption) and even operando/in-situ characterizations under electrolyte conditions. For electrocatalysis, identifying or designing MOFs that remain stable across a wide or targeted pH range is crucial, as this minimizes extrinsic interference and enables reliable interpretation of structural evolution and corresponding catalytic mechanisms.

Reviewer #3

General comment. In this resubmitted version, the authors have explained the peer review comments accordingly and improved the manuscript. The quality of this article has been improved significantly and is suitable for publication. So I recommend the direct acceptance of this manuscript.

Response: We sincerely appreciate your thoughtful evaluation of our revised manuscript. We are grateful for your recognition of the improvements made in response to the reviewers' comments. Thank you for your encouraging recommendation and supportive feedback.

References

1. Fang, Z.; Bueken, B.; De Vos, D. E.; Fischer, R. A., Defect-Engineered Metal–Organic Frameworks. *Angewandte Chemie International Edition* **2015**, *54* (25), 7234-7254.
2. Qiu, X.; Wang, R., From construction strategies to applications: Multifunctional defective metal-organic frameworks. *Coordination Chemistry Reviews* **2025**, *526*, 216356.
3. Xiang, W.; Zhang, Y.; Chen, Y.; Liu, C.-j.; Tu, X., Synthesis, characterization and application of defective metal–organic frameworks: current status and perspectives. *Journal of Materials Chemistry A* **2020**, *8* (41), 21526-21546.
4. Huang, Z.; Rath, J.; Zhou, Q.; Cherevan, A.; Naghdi, S.; Eder, D., Hierarchically Micro- and Mesoporous Zeolitic Imidazolate Frameworks Through Selective Ligand Removal. *Small* **2024**, *20* (21), 2307981.
5. Wan, G.; Sun, C.-J.; Freeland, J. W.; Fong, D. D., Defect-Driven Oxide Transformations and the Electrochemical Interphase. *Acc. Chem. Res.* **2021**, *54* (15), 3039-3049.
6. Portillo-Vélez, N. S.; Obeso, J. L.; de los Reyes, J. A.; Peralta, R. A.; Ibarra, I. A.; Huxley, M. T., Benefits and complexity of defects in metal-organic frameworks. *Communications Materials* **2024**, *5* (1), 247.
7. Ayala, P.; Naghdi, S.; Nandan, S. P.; Myakala, S. N.; Rath, J.; Saito, H.; Guggenberger, P.; Lakhanlal, L.; Kleitz, F.; Toroker, M. C.; Cherevan, A.; Eder, D., The Emergence of 2D Building Units in Metal-Organic Frameworks for Photocatalytic Hydrogen Evolution: A Case Study with COK-47. *Advanced Energy Materials* **2023**, *13* (31), 2300961.
8. Huang, Z.; Wang, Z.; Zhou, Q.; Rabl, H.; Naghdi, S.; Yang, Z.; Eder, D., Engineering of HO–Zn–N₂ Active Sites in Zeolitic Imidazolate Frameworks for Enhanced (Photo)Electrocatalytic Hydrogen Evolution. *Angewandte Chemie International Edition* **2025**, *64* (7), e202419913.
9. Zou, Z.; Wang, T.; Zhao, X.; Jiang, W.-J.; Pan, H.; Gao, D.; Xu, C., Expediting in-Situ Electrochemical Activation of Two-Dimensional Metal–Organic Frameworks for Enhanced OER Intrinsic Activity by Iron Incorporation. *ACS Catalysis* **2019**, *9* (8), 7356-7364.
10. Wang, Y.; Zhao, L.; Ma, J.; Zhang, J., Confined interface transformation of metal–organic frameworks for highly efficient oxygen evolution reactions. *Energy Environ. Sci.* **2022**, *15* (9), 3830-3841.
11. Wan, G.; Freeland, J. W.; Kloppenburg, J.; Petretto, G.; Nelson, J. N.; Kuo, D.-Y.; Sun, C.-J.; Wen, J.; Diulus, J. T.; Herman, G. S.; Dong, Y.; Kou, R.; Sun, J.; Chen, S.; Shen, K. M.; Schlom, D. G.; Rignanese, G.-M.; Hautier, G.; Fong, D. D.; Feng, Z.; Zhou, H.; Suntivich, J., Amorphization mechanism of SrIrO₃ electrocatalyst: How oxygen redox initiates ionic diffusion and structural reorganization. *Science Advances* *7* (2), eabc7323.
12. Che, Q.; van den Bosch, I. C. G.; Le, P. T. P.; Lazemi, M.; van der Minne, E.; Birkhölzer, Y. A.; Nunnenkamp, M.; Peerlings, M. L. J.; Safonova, O. V.; Nachtegaal, M.; Koster, G.; Baeumer, C.; de Jongh, P.; de Groot, F. M. F., In Situ X-ray Absorption Spectroscopy of LaFeO₃ and LaFeO₃/LaNiO₃ Thin Films in the Electrocatalytic Oxygen Evolution Reaction. *J. Phys. Chem. C* **2024**, *128* (13), 5515-5523.
13. Wan, X.; Liu, Q.; Liu, J.; Liu, S.; Liu, X.; Zheng, L.; Shang, J.; Yu, R.; Shui, J., Iron atom–cluster interactions increase activity and improve durability in Fe–N–C fuel cells. *Nature Communications* **2022**, *13* (1), 2963.

14. Pramanik, B.; Sahoo, R.; Das, M. C., pH-stable MOFs: Design principles and applications. *Coordination Chemistry Reviews* **2023**, *493*, 215301.
15. An, Y.; Lv, X.; Jiang, W.; Wang, L.; Shi, Y.; Hang, X.; Pang, H., The stability of MOFs in aqueous solutions—research progress and prospects. *Green Chemical Engineering* **2024**, *5* (2), 187-204.
16. Leus, K.; Bogaerts, T.; De Decker, J.; Depauw, H.; Hendrickx, K.; Vrielinck, H.; Van Speybroeck, V.; Van Der Voort, P., Systematic study of the chemical and hydrothermal stability of selected “stable” Metal Organic Frameworks. *Microporous and Mesoporous Materials* **2016**, *226*, 110-116.
17. Bai, Y.; Dou, Y.; Xie, L.-H.; Rutledge, W.; Li, J.-R.; Zhou, H.-C., Zr-based metal–organic frameworks: design, synthesis, structure, and applications. *Chemical Society Reviews* **2016**, *45* (8), 2327-2367.
18. Rath, B. B.; Vittal, J. J., Water Stable Zn(II) Metal–Organic Framework as a Selective and Sensitive Luminescent Probe for Fe(III) and Chromate Ions. *Inorganic Chemistry* **2020**, *59* (13), 8818-8826.
19. Zhan, Z.; Jia, Y.; Li, D.; Zhang, X.; Hu, M., A water-stable terbium-MOF sensor for the selective, sensitive, and recyclable detection of Al³⁺ and CO₃²⁻ ions. *Dalton Transactions* **2019**, *48* (40), 15255-15262.
20. Wang, K.; Huang, H.; Zhou, X.; Wang, Q.; Li, G.; Shen, H.; She, Y.; Zhong, C., Highly Chemically Stable MOFs with Trifluoromethyl Groups: Effect of Position of Trifluoromethyl Groups on Chemical Stability. *Inorganic Chemistry* **2019**, *58* (9), 5725-5732.
21. Liu, T.-F.; Zou, L.; Feng, D.; Chen, Y.-P.; Fordham, S.; Wang, X.; Liu, Y.; Zhou, H.-C., Stepwise Synthesis of Robust Metal–Organic Frameworks via Postsynthetic Metathesis and Oxidation of Metal Nodes in a Single-Crystal to Single-Crystal Transformation. *Journal of the American Chemical Society* **2014**, *136* (22), 7813-7816.